FERMILAB-PUB-23-641-T, MCNET-23-18

# A Portable Parton-Level Event Generator
# for the High-Luminosity LHC

**Enrico Bothmann[1]\*, Taylor Childers[2], Walter Giele[3], Stefan Höche[3], Joshua Isaacson[3]**
**and Max Knobbe[1]**

**1** Institut für Theoretische Physik, Georg-August-Universität Göttingen, 37077 Göttingen, Germany
**2** Argonne National Laboratory, Lemont, IL, 60439, USA
**3** Fermi National Accelerator Laboratory, Batavia, IL 60510, USA

\* enrico.bothmann@uni-goettingen.de

## Abstract

**The rapid deployment of computing hardware different from the traditional CPU+RAM model in data centers around the world mandates a change in the design of event generators for the Large Hadron Collider, in order to provide economically and ecologically sustainable simulations for the high-luminosity era of the LHC. Parton-level event generation is one of the most computationally demanding parts of the simulation and is therefore a prime target for improvements. We present a production-ready leading-order parton-level event generation framework capable of utilizing most modern hardware and discuss its performance in the standard candle processes of vector boson and top-quark pair production with up to five additional jets.**

# 1  Introduction

Monte-Carlo simulations of detector events are a cornerstone of high-energy physics [1, 2]. Simulated events are usually fully differential in momentum and flavor space, such that the same analysis pipeline can be used for both the experimental data and the theoretical prediction. This methodology has enabled precise tests of the Standard Model and searches for theories beyond it. Traditionally, the computing performance of event simulations has been of relatively little importance, due to the far greater demands of the subsequent detector simulation. However, the high luminosity of the LHC and the excellent understanding of the ATLAS and CMS detectors call for ever more precise calculations. This is rapidly leading to a situation where poor performance of event simulations can become a limiting factor for the success of experimental analyses [3–5], especially in view of modern approaches to detector simulation and reconstruction [6–10]. In addition, the impact of LHC computing on climate change must be considered, and its carbon footprint be kept as low as reasonably achievable [11]. All high-performance computing systems that are sufficiently large for the needs of the high-luminosity LHC consist of a mixture of CPU and GPU architectures. Therefore, any first step in minimizing the carbon footprint of theory predictions will require fully utilizing these systems through the development of CPU and GPU algorithms. Furthermore, a portable code would allow for the quick adoption of new more efficient hardware with minimal additional overhead from validation of the tools.

Alleviating this problem has been a focus of interest recently. Tremendous improvements of existing code bases have been obtained from improved event generation algorithms and PDF interpolation routines [12, 13]. Phase-space integrators have been equipped with adaptive algorithms based on neural networks [14–19] and updated with simple, yet efficient analytic solutions [20]. Negative weights in NLO matching procedures were reduced systematically [21, 22], and analytic calculations have been used to replace numerical methods when available [23]. Neural network based methods have been devised to construct surrogate matrix elements with improved numerical performance [24–27]. All those developments have in common that they operate on well-established code bases for the computation of matrix elements in the traditional CPU+RAM model. At the same time, the deployment of heterogeneous

computing systems is steadily accelerated by the increasing need to provide platforms for AI software. This presents a formidable challenge to the high-energy physics community, with its rigid code structures and slow-paced development, caused by a persistent lack of human resources. The Generator Working Group of the HEP Software Foundation and the HEP Center for Computational Excellence have therefore both identified the construction of portable simulation programs as essential for the success of the high-luminosity LHC. Some exploratory work was performed in [28–31], and first concrete steps towards generic solutions on modern GPUs have been reported in [32–38].

In this manuscript we will describe the hardware agnostic implementation of a complete leading-order parton-level event generator for processes that are simulated at high statistical precision at the LHC, including $Z/\gamma$+jets, $t\bar{t}$+jets, pure jets and multiple heavy quark production. Delivering such a framework first at leading order is a natural starting point, and addresses the computationally most expensive components of state-of-the-art multi-jet merged LHC simulations [13, 39]. The most significant obstacle in these computations is the low unweighting efficiency at high particle multiplicity, combined with exponential scaling (in the best case) of the integrand [39]. We demonstrate that new algorithms and widely available modern hardware alleviate this problem to a degree that renders previously highly challenging computations accessible for everyday analyses. We make the code base available for public use and provide an interface to a recently completed particle-level event simulation framework [40], enabling state-of-the art collider phenomenology with our new generator.

This manuscript is organized as follows. Section 2 recalls the main results of previous studies on matrix-element and phase-space generation and details the extension to a production-level code base. Section 3 introduces our new simulation framework, which we call PEPPER[1], and discusses the techniques for managing partonic sub-processes, helicity integration, projection onto leading-color configurations, and other aspects relevant for practical applications. Section 5 is focused on the computing performance of the new framework, both in comparison to existing numerical simulations and in comparison between different hardware. Section 6 presents possible future directions of development.

## 2 Basic algorithms

The computation of tree-level matrix elements and the generation of phase-space points are the components with the largest footprint in state-of-the-art LHC simulations [13, 39]. We aim to reduce their evaluation time as much as possible, while also focusing on simplicity, portability and scalability of the implementation. Reference [32] presented a dedicated study to find the best algorithms for the computation of all-gluon amplitudes on GPUs. Here we extend the method to processes including quarks. We also recall the results of Ref. [20], which presented an efficient concept for phase-space integration easily extensible to GPUs. We make a few simple changes in the algorithm, which lead to a large increase in efficiency without complicating the structure of the code or impairing portability.

### 2.1 Tree-level amplitudes

We use the all-gluon process often discussed in the literature as an example to introduce the basic concepts of matrix-element computation. A color and spin summed squared $n$-gluon

---

[1]PEPPER is an acronym for Portable Engine for the Production of Parton-level Event Records.
The PEPPER source code is available at https://gitlab.com/spice-mc/pepper.

tree-level amplitude is defined as

$$
|\mathcal{A}(1,\dots,n)|^2 = \sum_{a_1\dots a_n} \mathcal{A}^{\mu_1\cdots\mu_n}_{a_1\dots a_n}(p_1,\dots,p_n)\left(\mathcal{A}^{\nu_1\cdots\nu_n}_{a_1\dots a_n}(p_1,\dots,p_n)\right)^\dagger \prod_{i=1}^n \sum_{\lambda_i} \epsilon^{\lambda_i}_{\mu_i}(p,k)\epsilon^{\lambda_i\,\dagger}_{\nu_i}(p,k)\,,
$$
(1)

where each gluon with label $i$ is characterized by its momentum $p_i$, its color $a_i$ and its helicity $\lambda_i = \pm$. The squared $n$-gluon amplitude then takes the form

$$
|\mathcal{A}(1,\dots,n)|^2 = \sum_{\lambda_1\dots\lambda_n} \sum_{a_1\dots a_n} \mathcal{A}^{\lambda_1\dots\lambda_n}_{a_1\dots a_n}(p_1,\dots,p_n)\left(\mathcal{A}^{\lambda_1\dots\lambda_n}_{a_1\dots a_n}(p_1,\dots,p_n)\right)^\dagger\,,
$$
(2)

where the $\mathcal{A}^\lambda$ are the chirality-dependent scattering amplitudes obtained by contracting $\mathcal{A}^\mu$ with the helicity eigenstates $\epsilon^\lambda_\mu$ of the external gluons. The treatment of color in this calculation is a complex problem. The most promising approach for LHC physics at low to medium jet multiplicity is to explicitly sum over color states using a color-decomposition with a minimal basis [32]. In the pure gluon case, this basis is given by the adjoint representation [41–43]

$$
\mathcal{A}^{\lambda_1\dots\lambda_n}_{a_1\dots a_n}(p_1,\dots,p_n) = \sum_{\vec{\sigma}\in S_{n-2}} (F^{a_{\sigma_2}}\dots F^{a_{\sigma_{n-1}}})_{a_1 a_n} A^{\lambda_1\dots\lambda_n}(p_1,p_{\sigma_2},\dots,p_{\sigma_{n-1}},p_n)\,,
$$
(3)

where $F^a_{bc} = if^{abc}$ and $f^{abc}$ are the SU(3) structure constants. The functions $A$ are called color-ordered or partial amplitudes and are stripped of all color information, which is now contained entirely in the prefactor. If the amplitudes carry helicity labels, they are often also referred to as helicity amplitudes. The multi-index $\vec{\sigma}$ runs over all permutations $S_{n-2}$ of the $(n-2)$ gluon indices $2,\dots,n-1$. The color basis thus defined is minimal and has $(n-2)!$ elements.

For amplitudes involving not only gluons but also quarks, the minimal color basis is given in terms of Dyck words [44,45]. A Dyck word is a set of opening and closing brackets, such that the number of opening brackets is always larger or equal to the number of closing ones for any subset starting at the beginning of the Dyck word. For example for four characters and one type of bracket, there are two Dyck words, (()) and ()(). In the context of QCD amplitude computations, every opening bracket represents a quark and every closing one an anti-quark. Different types of brackets can appear and indicate differently flavored quarks. Similar to the gluon case in Eq. (3), one may keep two parton indices fixed throughout the computation and permute all others, as long as the permutation forms a valid Dyck word. The number of partial amplitudes in this basis is minimal. For $n$ particles and $k$ distinct quark pairs, it is given by $(n-2)!/k!$, which is a generalization of the minimal all-gluon ($k=0$) result. The color factors needed for the computations can be evaluated using the algorithm described in [46,47].

There are various algorithms to compute the partial amplitudes $A(p_1,\dots,p_n)$ in Eq. (3) (we now omit helicity labels for brevity). The numerical efficiency of the most promising approaches has been compared in [32,48–50]. It was found that, for generic helicity configurations and arbitrary particle multiplicity, the Berends–Giele recursive relations [51–53] offer the best performance. We therefore choose this method for our new matrix-element generator. The basic objects in the Berends–Giele approach are off-shell currents, $J(1,\dots,n)$. In the case of an all-gluon amplitude, they are defined as

$$
J_\mu(1,2,\dots,n) = \frac{-ig_{\mu\nu}}{p_{1,n}^2}\Bigg\{ \sum_{k=1}^{n-1} V_3^{\nu\kappa\lambda}(p_{1,k},p_{k+1,n})J_\kappa(1,\dots,k)J_\lambda(k+1,\dots,n)
$$
$$
+ \sum_{j=1}^{n-2}\sum_{k=j+1}^{n-1} V_4^{\nu\rho\kappa\lambda}J_\rho(1,\dots,j)J_\kappa(j+1,\dots,k)J_\lambda(k+1,\dots,n)\Bigg\}\,,
$$
(4)

where the $p_i$ denote the momenta of the gluons, $p_{i,j} = p_i + \ldots + p_j$ and $V_3^{\nu\kappa\lambda}$ and $V_4^{\nu\rho\kappa\lambda}$ are the color-ordered three- and four-gluon vertices:

$$
\begin{aligned}
V_3^{\nu\kappa\lambda}(p,q) &= i\,\frac{g_s}{\sqrt{2}}\left( g^{\kappa\lambda}(p-q)^\nu + g^{\lambda\nu}(2q+p)^\kappa - g^{\nu\kappa}(2p+q)^\lambda \right), \\
V_4^{\nu\rho\kappa\lambda} &= i\,\frac{g_s^2}{2}\left( 2g^{\nu\kappa}g^{\rho\lambda} - g^{\nu\rho}g^{\kappa\lambda} - g^{\nu\lambda}g^{\rho\kappa} \right).
\end{aligned}
\tag{5}
$$

The external particle currents, $J_\mu(i)$, are given by the helicity eigenstates, $\epsilon_\mu(p_i)$. The complete amplitude $A(p_1,\ldots,p_n)$ is obtained by putting the $(n-1)$-particle off-shell current $J_\mu(1,\ldots,n-1)$ on-shell and contracting it with the external polarization $J_\mu(n)$:

$$
A(p_1,\ldots,p_n) = J_\mu(n)\,p_{1,n}^2\,J^\mu(1,\ldots,n-1)\,.
\tag{6}
$$

A major advantage of this formulation of the calculation is that it can straightforwardly be extended to processes including massless and massive quarks, as well as to calculations involving non-QCD particles. The external currents for fermions are given by spinors, and the three- and four-particle vertices are fully determined by the Standard Model interactions.

## 2.2 Phase-space integration

The differential phase space element for an $n$-particle final state at a hadron collider with fixed incoming momenta $p_a$ and $p_b$ and outgoing momenta $\{p_1,\ldots,p_n\}$ is given by [54]

$$
\mathrm{d}\Phi_n(a,b;1,\ldots,n) = \left[ \prod_{i=1}^n \frac{\mathrm{d}^3\vec{p}_i}{(2\pi)^3\,2E_i} \right] (2\pi)^4 \delta^{(4)}\!\left( p_a + p_b - \sum_{i=1}^n p_i \right).
\tag{7}
$$

For processes without $s$-channel resonances, it is convenient to parameterize Eq. (7) by

$$
\mathrm{d}x_a\,\mathrm{d}x_b\,\mathrm{d}\Phi_n(a,b;1,\ldots,n) = \frac{2\pi}{s}\left[ \prod_{i=1}^{n-1} \frac{1}{16\pi^2}\,\mathrm{d}p_{i,\perp}^2\,\mathrm{d}y_i\,\frac{\mathrm{d}\phi_i}{2\pi} \right] \mathrm{d}y_n\,,
\tag{8}
$$

where, $p_{i,\perp}$, $y_i$ and $\phi_i$ are the transverse momentum, rapidity and azimuthal angle of momentum $i$ in the laboratory frame, and where $x_a$ and $x_b$ are the Bjørken variables of the incoming partons. In processes with unambiguous $s$-channel topologies, such as Drell–Yan lepton pair production, one may instead use the strategy of [20] and parameterize the decay of the resonance using the well-known two-body decay formula

$$
\mathrm{d}\Phi_2(\{1,2\};1,2) = \frac{1}{16\pi^2}\,\frac{\sqrt{(p_1 p_2)^2 - p_1^2 p_2^2}}{(p_1+p_2)^2}\,\mathrm{d}\cos\theta_1^{\{1,2\}}\,\mathrm{d}\phi_1^{\{1,2\}}\,,
\tag{9}
$$

which, as written here, has been evaluated in the center-of-mass frame of the decaying particle. The $t$- and $s$-channel building blocks in Eqs. (8) and (9) can be combined using the standard factorization formula [55]

$$
\mathrm{d}\Phi_n(a,b;1,\ldots,n) = \mathrm{d}\Phi_{n-m+1}(a,b;\{1,..,m\},m+1,\ldots,n)\,\frac{\mathrm{d}s_{\{1,..,m\}}}{2\pi}\,\mathrm{d}\Phi_m(\{1,..,m\};1,\ldots,m)\,.
\tag{10}
$$

We will refer to this minimal integration technique as the basic CHILI method. It is both simple to implement, and reasonably efficient due to the compact form of the compute kernels [20]. The latter aspect is especially important in the context of code portability and maintenance.

Compared to the first implementation in Ref. [20], we apply two improvements which increase the integration efficiency significantly: 1) In the azimuthal angle integration of Eq. (8),

the sum of previously generated momenta serves to define $\phi = 0$. 2) Assuming that particles 1 through $m$ are subject to transverse momentum cuts, and assuming $n$ final-state particles overall, we generate the transverse momenta of the $n-m$ particles not subject to cuts according to a peaked distribution given by $1/(p_{i,\perp} + p_{\perp,0})$, where $p_{\perp,0} = |\sum_{i=1}^{m} \vec{p}_{i,\perp}|/(n-m)$. We also use $\sum_{i=1}^{m} \vec{p}_{i,\perp}$ to define $\phi = 0$ for the corresponding azimuthal angle integration.

# 3 Event generation framework

The construction of a production-ready matrix element generator requires many design choices beyond the basic matrix element and phase space calculation routines discussed in Sec. 2. In most experimental analyses, flavors are not resolved inside jets, and the sum over partons can be performed with the help of symmetries among the QCD amplitudes. In addition, an efficient strategy must be found to perform helicity sums. For a parallelized code such as PEPPER, two additional questions must be addressed: how to arrange the event data for efficient calculations on massively parallel architectures, and how to write the generated parton-level events to persistent storage without unnecessarily limiting the data transfer rate. We will discuss these aspects in the following.

## 3.1 Summation of partonic channels and running coupling

In the PEPPER event generator, the partonic processes which contribute to the hadronic cross section are arranged into groups, such that all processes within a group have the same partonic matrix element. For any given phase-space point, the matrix element squared is evaluated only once, and then multiplied by the running strong coupling times the sum over the product of partonic fluxes, given by $\alpha_s(\mu_R^2) \sum_{\{i,j\}} f_i(x_1, \mu_F^2) f_j(x_2, \mu_F^2)$. Here, the indices $\{i, j\}$ run over all incoming parton pairs that contribute to the group, and the $\mu_{R,F}^2$ are the renormalization and the factorization scale, respectively, which can be evaluated dynamically in PEPPER, i.e. as various functions of the external momenta. The strong coupling $\alpha_s$ and the PDF values $f_i$ are evaluated using a modified version of the LHAPDF v6.5.4 library [56], that supports parallel evaluation on various architectures via CUDA and Kokkos; this will be further discussed in Sec. 5.

As an example, for the $pp \to t\bar{t} + j$ process, considering all quarks but the top quark to be massless, three partonic subprocess groups are identified: $q\bar{q} \to t\bar{t}g$ (5 subprocesses: $d\bar{d} \to t\bar{t}g$, $u\bar{u} \to t\bar{t}g$, ...), $gq \to t\bar{t}q$ (10 subprocesses: $gd \to t\bar{t}d$, $g\bar{d} \to t\bar{t}\bar{d}$, $gu \to t\bar{t}u$, $g\bar{u} \to t\bar{t}\bar{u}$ ...), and $gg \to t\bar{t}g$ (1 subprocess). When helicities are explicitly summed over, each group of partonic subprocesses contributes one channel to the multi-channel Monte Carlo used to handle the group. When helicities are Monte-Carlo sampled, one channel is used for each non-vanishing helicity configuration.

In order to produce events with unambiguous flavour structure, which is necessary for further simulation steps such as parton showering and hadronization, one of the partonic subprocesses of the group is selected probabilistically according to its relative contribution to the sum over the product of partonic fluxes.

## 3.2 Helicity integration

The PEPPER event generator provides two options to perform the helicity sum in Eq. (2). One is to explicitly sum over all possible external helicity states, the other is to perform the sum in a Monte-Carlo fashion. In both cases, exactly vanishing helicity amplitudes are identified at the time of initialization and removed from the calculation.

The two methods offer different advantages. Summing helicity configurations explicitly reduces the variance of the integral, but the longer evaluation time can lead to a slower overall convergence, especially for higher final-state multiplicities. To improve the convergence when helicity sampling is used, we adjust the selection weights during the initial optimization phase with a multi-channel approach [57]. This optimization is particularly important in low multiplicity processes with strong hierarchies among the non-vanishing helicity configurations.

For helicity-summed amplitudes we implement two additional optimizations. Considering Eq. (6) we observe that a change in the helicity of $J_\mu(n)$ does not require a recomputation of the remaining Berends–Giele currents, and we can efficiently calculate two helicity configurations at once. Furthermore, for pure QCD amplitudes, we make use of their symmetry under the exchange of all external helicities [58], again reducing the number of independent helicity states by a factor of two.

### 3.3 Event data layout and parallel event generation

In contrast to most traditional event generators, which produce only a single event at any given time, PEPPER generates events in batches, which enables the parallelized evaluation of all events in a batch on multithreaded architectures. In the first step, all phase-space points and weights for the event batch are generated. Then the Berends–Giele recursion is run for all events in the batch to calculate the matrix elements, etc., and finally the accepted events are aggregated for further processing and output.

To ensure data locality and hence good cache efficiency for this approach, any given property of the event is stored contiguously in memory for the entire batch. For example, the $x$ components of the momenta of a given particle are stored in a single array. This kind of layout is often called a struct-of-arrays (SoA), as opposed to an array-of-structs (AoS) layout, for which the data of all properties of an individual event would be laid out contiguously instead.

We have tested that the SoA layout is not only required to achieve peak performance on GPU-like hardware, but also that it does not degrade performance when running serially on a single CPU thread. We expect this to be the case, given that the code is organised as a pipeline of relatively simple compute kernels that operate on the common event batch data. Since the individual kernels are kept simple, they often only operate on a subset of the event data (e.g. only on the external momenta, or they only update the event weight, etc.) Thus, the SoA layout allows the CPU to cache locally relevant event data only (instead of caching all data of a single event, including data for properties that are not being used by a given algorithm).

In App. D, we show CPU results for $pp \to t\bar{t} + 3$ jets, comparing SoA with AoS performance results with different event batch sizes, which is a configurable runtime parameter in PEPPER. Both layouts profit from increasing the batch size to 10 or 100 events, compared with processing single-event batches, but the speed-ups are moderate with about 10 %. For more details, see App. D. We also find that the SoA performance is on-par with the AoS performance. As expected, we find more significant speed-ups with increased batch size when considering much simpler processes, such as $d\bar{d} \to u\bar{u}$. Here, using a batch size of 100 events yields a speed-up of about a factor 3 instead, which makes sense given the small amount of data per event. Also in this case, we find that the SoA layout performs as well as the AoS layout.

The main reason for choosing an SoA layout and batched processing is to parallelize the generation of the events of an entire batch on many-threaded parallel processing units such as GPUs, by generating one event per thread. In this case, the contiguous layout of the data for each event property ensures that also data reading and writing benefits from available hardware parallelism. Even after this optimization, we find that the algorithm for the matrix element evaluation described in Sec. 2 is memory-bound rather than compute-bound, i.e. the bottleneck for the throughput is the speed at which lines of memory can be delivered to/from the processing units. We address this by reducing the reads/writes as much as possible. For

example, we use the fact that massless spinors can be represented by only two components to halve the number of reads/writes for such objects.

Another consideration is the concept of branch divergence on common GPU hardware. The threads are arranged in groups, and the threads within each of the groups are operating in lock-step[2], i.e. ideally a given instruction is performed for all of these threads at the same time. However, if some of the threads within the group have diverged from the others by taking another branch in the program, they have to wait for the others until the execution branches merge again. Hence, to maximize performance, it is important to prevent branch divergence as much as feasible. We do so by selecting the same partonic process group and helicity configuration for groups of 32 threads/events within a given event batch.[3] This implies that events of the same group that are accepted and then written to storage will be correlated. Other than in the most trivial setups, for unweighted event generation the low efficiencies imply that most accepted events are the only event in their group that are accepted, thus removing this correlation almost entirely. In post-processing, this correlation can be removed if necessary by a random shuffling of the events.[4] However, in most applications the ordering within a sample is not relevant, provided that the entire sample is processed, which should consist of a large number of such blocks.

With this data structure and lock-step event generation in place, PEPPER achieves a high degree of parallelization. The parallelized parts of the event generation include the following steps:

1. Generate random numbers.

2. Generate external momenta with an optional phase-space bias.

3. Apply phase-space cuts (i.e. set the weight of an event to zero if its external momenta do not pass the cuts).

4. Evaluate phase-space sampling weight.

5. Evaluate dynamical unphysical renormalization and factorization scales $\mu_{R,F}$.

6. Evaluate the running coupling $\alpha_S(\mu_R)$, the sum over initial states $i, j$ and the corresponding partonic fluxes $f_i(x_1, Q^2)f_j(x_2, Q^2)$.

7. Sample (or sum) helicities, evaluate helicity sampling weight and calculate external polarization vectors.

8. Evaluate amplitudes recursively and sum the squared amplitudes over color configurations.

9. Unweight events against the weight maximum (set event weight to zero if the event is rejected).

10. Optionally, project onto a leading color configuration.

11. Copy non-zero events from the device to the host.

At this point, non-zero events are written to storage, which is discussed in Sec. 3.5. Furthermore, note that while we can skip events that do not pass the phase-space cuts (see step 3) easily during serial processing, for parallel processing we will either have threads with non-passing events doing idle work or wait, or we keep generating events until we have accepted a sufficient

---

[2]A group of threads operating in lock-step is often called a "warp", and usually consists of 32 threads.

[3]This group size is configurable at compile time. The default of 32 is chosen to match the GPU warp size.

[4]Such a shuffling tool for LHEH5 files can be found at https://gitlab.com/spice-mc/tools/lheh5_shuffle.

number before proceeding, which would include sort and copy operations. Let us call these two methods the "take-the-hit" and the "enrichment" method. Which one is more efficient will ultimately depend on the phase-space efficiency, i.e. the relative number of events passing the cuts. We find that for the $pp \to Z + 5$ jets and $pp \to t\bar{t} + 4$ jets processes, with the standard cuts defined in Eq. (11), our phase-space efficiencies after optimisation are at 86 % and 90 %, respectively, such that even an ideal "enrichment" method could not increase the throughput significantly. We therefore choose to use the trivially implemented "take-the-hit" method.

## 3.4 Portability solutions

To make our new event generator suitable for usage on a wide variety of hardware platforms, we use the Kokkos portability framework [59, 60], which allows a single and therefore easily maintainable source code to be used for a multitude of architectures. Furthermore, Kokkos automatically performs architecture dependent optimizations e.g. for memory alignment and parallelization; C++ classes are provided to abstract data representations and facilitate data handling across hardware. This data handling needs to be efficient both for heterogeneous (CPU and GPU) as well as homogeneous (CPU-only) architectures, hence we carefully ensure that no unnecessary memory copies are made. The code is structured into cleanly delineated computational kernels, thus separating (serial) organizational parts and (parallel) actual computations. This design facilitates optimal computing performance on different architectures.

The PEPPER v1.0.0 release also includes variants for conventional sequential CPU evaluation and CUDA-accelerated evaluation on Nvidia GPU. Early versions for some components of these codes were first presented for the gluon scattering case in [32]. The main Kokkos variant is modeled on the CUDA variant. This allowed us to do continuous cross-checks of the physics results and the performance between the variants during the development. All variants support the Message Passing Interface (MPI) standard [61] to execute in parallel across many cores.

We also developed the CUDA and Kokkos version of the PDF interpolation library LHAPDF. To evaluate PDF values, LHAPDF performs cubic interpolations on precomputed grids supplied by PDF fitting groups. Furthermore, the PDF grids are usually supplied with a corresponding grid for the strong coupling constant, $\alpha_s$. The strong coupling constant and the PDF are core ingredients of any parton-level event simulation, and the extensive use of LHAPDF justifies porting them along with the event generator. Speed-ups are achieved by the parallel evaluation and reduction of memory copies required between the GPU and the CPU. Our port of LHAPDF focuses on the compute intensive interpolation component, and leaves all the remaining components untouched. The resulting code will be made available in a future public release of LHAPDF.

## 3.5 Event output

The generated (and unweighted) events must eventually be written to disk (or passed on to another program via a UNIX pipe or a more direct program interface). When generating events on a device with its own memory, the event information (momenta, weights, . . . ) must first be copied from the device memory to the host memory. Since the event generation rate on the device can be very large, the transfer rate is an important variable. Fortunately, there is no need to output events which did not pass phase-space cuts or were rejected by the unweighting. Hence, PEPPER filters out these zero events on the device and then transfers only non-zero events to the host. This leads to event transfer rates that can be orders of magnitude smaller than if all data were transferred, especially for low phase-space and/or unweighting efficiencies.

We project the events to a leading color configuration in order to store a valid color configuration that can be used for parton showers. To that end, we compute the leading-color factors corresponding to the full-color point as described in Sec. 2. We then stochastically

select a permutation of external particles among the leading color amplitudes and use this permutation to define a valid color configuration.

The user can choose to let PEPPER output events in one of three standard formats:

**ASCII v3** This is the native plain text format of the HepMC3 Event Record Library [62]. The events can be written to an (optionally compressed) file or to standard output. The format is useful for direct analysis of the parton-level events with the Rivet library [63], for example. There is no native MPI support, such that events are written to separate files for each MPI rank.

**LHEF v3** This is the XML-based LHE file format [64] in its version 3.0 [65], which is widely used to encode parton-level events for further processing, e.g. using a parton-shower program. The events can be written to an (optionally compressed) file or to standard output. There is no native MPI support, such that events are written to separate files for each MPI rank.

**LHEH5** This is an HDF5 database library [66] based encoding of parton-level events. HDF5 is accessed through the HighFive header library [67]. While the format contains mostly the same parton-level information as an LHEF file, its rigid structure and HDF5's native support for collective MPI-based writing makes its use highly efficient in massively parallel event generation [40]. HDF5 supports MPI, such that events are written to a single file even in a run with multiple MPI ranks. LHEH5 files can be processed with Sherpa [68,69] and Pythia [70,71]. The LHEH5 format and the existing LHEH5 event generation frameworks have been described in [39,40].

## 4  Validation

To validate the implementation of our new generator, we compare PEPPER v1.0.0 with SHERPA v2.3.0's [69] internal matrix element generator AMEGIC [72]. In both cases we further process the parton-level events using SHERPA's default particle-level simulation modules for the parton shower [73], the cluster hadronization [74] and Sjöstrand–Zijl-like [75] multiple parton interactions (MPI) model [76]. In the case of AMEGIC, the entire event processing is handled within the SHERPA event generator, while in the case of PEPPER, the parton-level events are first stored in the LHEH5 format, and then read by SHERPA for the particle-level simulation, as described in Sec. 3.5 and Ref. [40].[5]

The first process we consider is $pp \to e^+e^- + n$ jets at $\mathcal{O}(\alpha_{\text{EW}}^2)$ with $n = 1,\ldots,4$. The center-of-mass energy of the collider is chosen to be $\sqrt{s} = 14\,\text{TeV}$. For the renormalization and factorization scales, we choose $\mu_R^2 = \mu_F^2 = \mu^2 = H_T'^2 = m_{\perp,e^+e^-}^2 + \sum_{i=1}^n p_{\perp,i}^2$, where $m_{\perp,e^+e^-}$ is the transverse mass of the dilepton system and $p_{\perp,i}$ is the transverse momentum of the $i$th final-state parton. We employ the following parton-level cuts:

$$p_{\perp,j} \geq 30\,\text{GeV}, \qquad |\eta_j| \leq 5.0, \qquad \Delta R_{jj} \geq 0.4, \qquad 66\,\text{GeV} \leq m_{e^+e^-} \leq 116\,\text{GeV}. \tag{11}$$

We use the NNPDF3.0 PDF set `NNPDF30_nlo_as_0118` [77] to parametrize the structure of the incoming protons, and the corresponding definition of the strong coupling, via the LHAPDF v6.5.4 library [56]. Particle-level events are passed to the Rivet analysis framework v3.1.8 [63]

---

[5]We choose here to include particle-level simulation steps in order to test not only the correctness of the calculation in PEPPER, but at the same time that the event files are written out correctly by PEPPER and read in and processed correctly by SHERPA. However, including the particle-level simulation might dilute deviations in the parton-level calculation. Therefore, in App. C, we repeat the validation at the parton level, without any additional simulation steps.

via the HepMC event record v3.2.6 [62]. The two observables we present are the $Z$ boson rapidity $y_Z$, and its transverse momentum $p_\perp^Z$ as defined in the `MC_ZINC` Rivet analysis.

The comparison results for these two observables are shown in Fig. 1. The smaller plots shown on the right display the deviations between the results from PEPPER + SHERPA and the ones from SHERPA standalone, normalized to the $1\sigma$ standard deviation of the SHERPA results. We observe agreement between the two predictions at the statistical level for all jet multiplicities.

To quantify the agreement, we test the null hypothesis that the deviations are distributed according to the standard normal using the Kolmogorov–Smirnov test [78–80]. We choose a confidence level of 95 %; that is, we reject the null hypothesis (*i.e.* that the two distributions are identical) in favor of the alternative if the $p$-value is less than 0.05. We find that all $p$-values are greater than 0.05. The individual $p$-values are quoted in Fig. 1 for each jet multiplicity.

The second process we consider is $pp \to t\bar{t} + n\,\text{jets}$ at $\mathcal{O}(\alpha_{\text{EW}}^0)$ with $n = 0, \ldots, 3$. The setup is identical to the one for Drell-Yan lepton pair production, except that the renormalization and factorization scales are defined by $\mu_R^2 = \mu_F^2 = \mu^2 = H_{T,M}^2 = m_{\perp,t}^2 + m_{\perp,\bar{t}}^2 + \sum_{i=1}^n p_{\perp,i}^2$. The top quark decay chain is performed by SHERPA in the narrow-width approximation as described in [69]. The observables used for the validation are the azimuthal angle $\Delta\phi$ between two light jets and the $H_T$ of all jets, as defined in the `MC_TTBAR` Rivet analysis in its semi-leptonic mode.

The comparison for these two observables is shown in Fig. 2. Again, the figure on the right shows the deviations of the PEPPER + SHERPA and the SHERPA standalone predictions for the different jet multiplicities, normalized to the $1\sigma$ uncertainty of the SHERPA predictions. We find agreement between the two results at the statistical level for all jet multiplicities, with the Kolmogorov–Smirnov test $p$-values all being greater than 0.05.

# 5   Performance

Performance benchmarks of generators for novel computing architectures in comparison to existing tools are not entirely straightforward. Differences in floating point performance, memory layout, bus performance, and other aspects of the hardware may bias any would-be one-on-one comparison. We therefore choose to analyze the code performance in two steps. First, we compare a single-threaded CPU version of PEPPER to one of the event generators used at the LHC, thus establishing a baseline. We then perform a cross-platform comparison of PEPPER itself. We thereby aim to demonstrate that the CPU performance of PEPPER is on par with the best available algorithms, and that the underlying code is a good implementation that can easily be ported to new architectures with reasonable results.

We note that the core algorithms of PEPPER (then referred to as BLOCKGEN) have been studied in detail in our gluon-only pathfinder study [32], and we refer the reader to this study for more technical performance results and comparisons with alternative algorithms. In addition, we provide PEPPER profiling data for $e^+e^- + n\,\text{jets}$ ($n = 0, \ldots, 4$) and $t\bar{t} + n\,\text{jets}$ ($n = 0, \ldots, 5$) production runs on an Nvidia V100 GPU for further study [81, 82]. However, in the following, we instead focus on high-level portability and performance comparisons on different architectures. We only repeat here that the performance of PEPPER is still memory bound on the GPU. In fact, the addition of fermions in PEPPER requires the use of complex-valued currents in the recursion, which further intensifies memory operations.

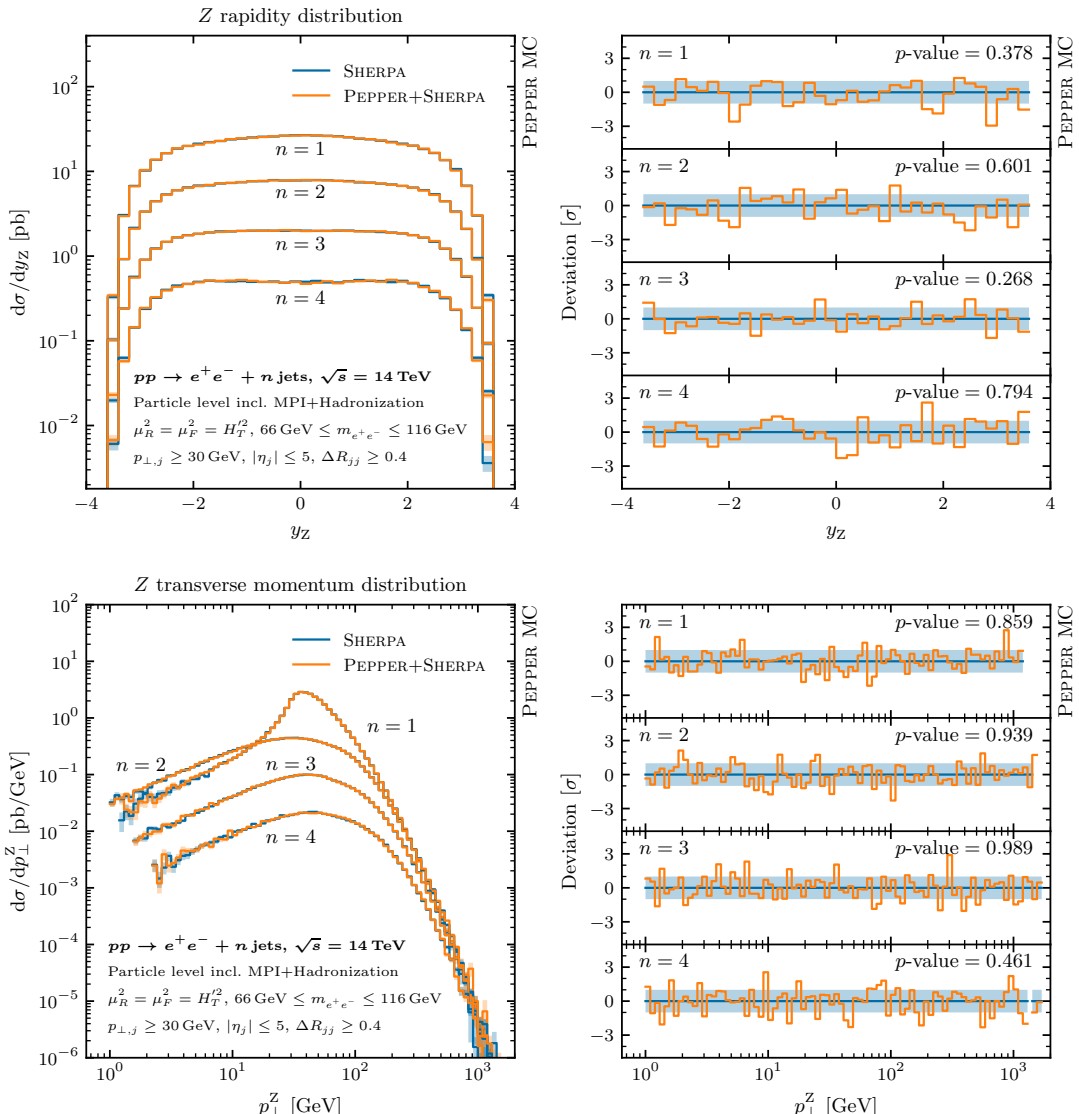

Figure 1: Particle-level validation for two observables for the $pp \to Z + n$ jets process, comparing SHERPA standalone results with PEPPER + SHERPA results, where the parton-level events are generated by PEPPER and then read in by SHERPA to perform the additional particle-level simulation. The left plots show the distributions, while the right plots show the deviations between SHERPA and PEPPER + SHERPA individually for each $n$, normalized to the $1\sigma$ standard deviation of the SHERPA result. For each $n$, the $p$-value of a Kolmogorov–Smirnov test is shown for the hypothesis that the deviations follow a standard normal distribution.

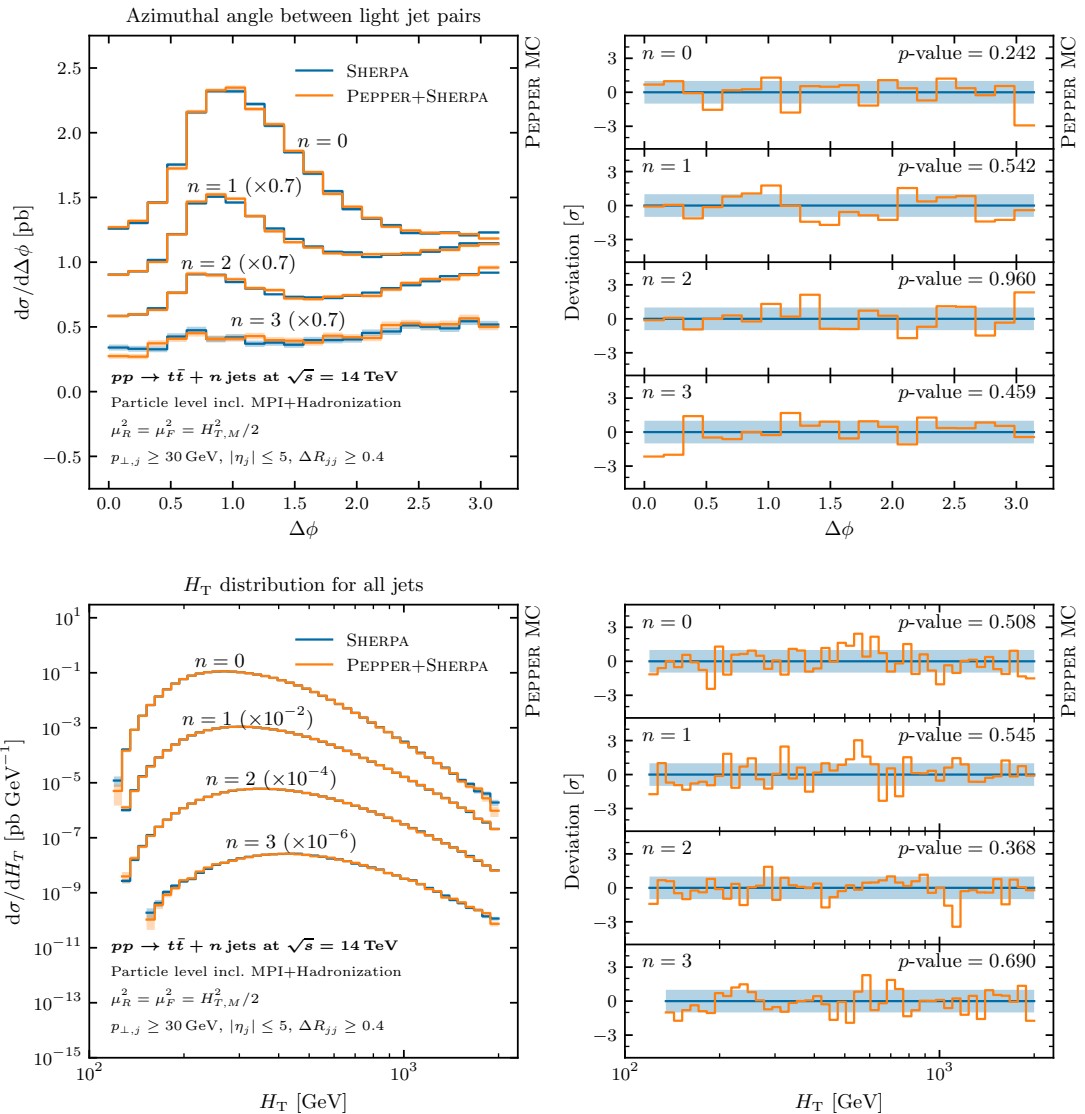

Figure 2: Particle-level validation for two observables of the $pp \to t\bar{t} + n$ jets process, comparing SHERPA standalone results with PEPPER + SHERPA results, for which the parton-level events are generated by PEPPER and then read in by SHERPA to perform the additional particle-level simulation steps. The left plots show the distributions, while the right plots show the deviations between SHERPA and PEPPER + SHERPA individually for each $n$, normalized to the $1\sigma$ standard deviation of the SHERPA result. For each $n$, the $p$-value of a Kolmogorov–Smirnov test is shown for the hypothesis that the deviations follow a standard normal distribution.

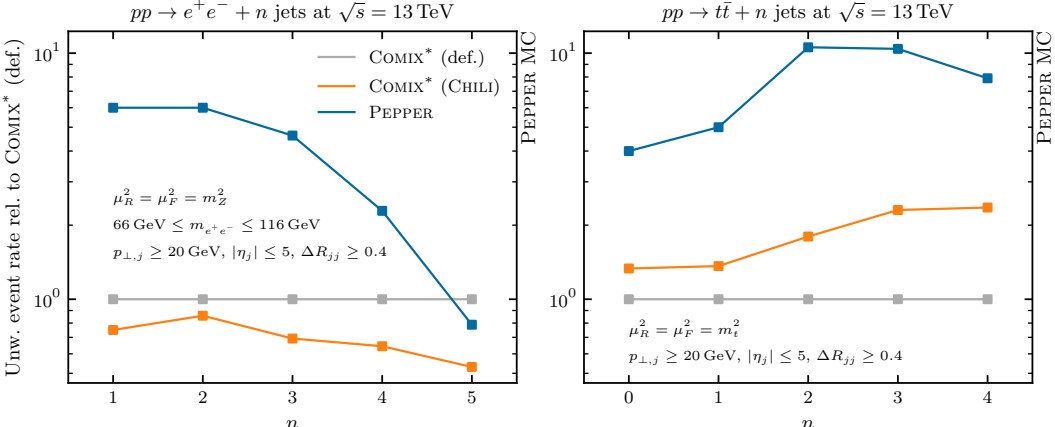

Figure 3: Event rates for generation and storage of unweighted events for $e^+e^- + n$ jets (left) and $t\bar{t} + n$ jets (right). We compare COMIX, combined with its default and with the basic CHILI phase-space generator, and PEPPER, normalized to the COMIX result. The asterisk in the COMIX label indicates that the generator is used in a non-default configuration, splitting the process sum into different Monte-Carlo channels, grouping by gluons, valence and sea quarks. The results were generated on a single core of an Intel Xeon E5-2650 v2 CPU. For details, see the main text.

## 5.1 Baseline CPU performance

Figure 3 presents a comparison of the CPU version of PEPPER and the parton-level event generator COMIX[6] [83]. We measure the rate at which unweighted events are generated and stored for $e^+e^- + n$ jets and $t\bar{t} + n$ jets production at the LHC. The center-of-mass energy of the collision is set to $\sqrt{s} = 13$ TeV, and the partons are required to fulfill $p_{T,j} > 20$ GeV, $|\eta_j| < 5$ and $\Delta R_{jj} > 0.4$. For $e^+e^- + n$ jets production, an additional cut of $66$ GeV $< m_{e^+e^-} < 116$ GeV is applied to the invariant mass of the dilepton system. For simplicity, the renormalization and factorization scales are fixed to $\mu_R = \mu_F = m_Z$ in $e^+e^- + n$ jets production, and to $\mu_R = \mu_F = m_t$ in $t\bar{t} + n$ jets production. For better comparability of the measurements, we show COMIX both in combination with its recursive phase space generator [83] and with the modified basic CHILI integrator [20] that is also used by PEPPER, cf. Sec. 2 for details. The raw data for Fig. 3 are listed in App. A.

For $e^+e^- + n$ jets production, we find that COMIX combined with CHILI yields a slightly reduced performance compared to COMIX combined with its default phase-space integrator. As observed in [20], the CHILI integrator has reduced unweighting efficiency in this case, but generates points much faster. Combining these factors results in the slightly reduced efficiency compared to the COMIX default. For PEPPER we observe a large throughput gain compared to COMIX, especially at low multiplicities, while COMIX only begins to outperform PEPPER for $n = 5$ additional jets. This nicely reflects the advantages of the different algorithms: The color-dressed recursion implemented in COMIX adds computational overhead compared to the explicit color sum in PEPPER, but the improved scaling takes over at some point. In the $t\bar{t}$+jets production, we observe a better performance of the CHILI integrator comparing the two COMIX results. This is also reflected in the initially increasing gain factor of PEPPER when compared to the default COMIX result. Despite the rather intricate color structure of the $t\bar{t}$+jets processes, PEPPER outperforms COMIX for all jet multiplicities tested here.

The results show that the single-threaded baseline performance of PEPPER is on par with that

---

[6]We use a non-default mode for COMIX, splitting the process into Monte-Carlo channels grouped by gluons, valence and sea quarks. This is similar to PEPPER's strategy and increases the efficiency of the event generation.

of an existing established generator like COMIX. The factorial scaling with multiplicity of the PEPPER algorithm has no adverse effects in the multiplicity range of interest, where PEPPER is in all cases as fast or faster than COMIX. Moreover, the efficiency of the basic CHILI phase-space generator used by PEPPER is on par with the much more complex recursive multi-channel phase-space generator implemented in COMIX, confirming earlier results [20].

## 5.2   Performance on different hardware

After establishing the baseline performance, we now study the suitability of PEPPER for different hardware architectures. A variety of computing platforms were used to measure the portability and performance. This list defines the architecture labels used in the figures that follow:

**2×Skylake8180**  Intel Xeon Platinum 8180M CPU at 2.50 GHz with 768 GB of memory. These are 28-core processors; and the machine contains two CPUs each. Our performance tests utilize all 56 cores unless otherwise noted, hence the "2×" in our label.

**V100**  Nvidia V100(SXM2) GPU with 32 GB of memory.

**A100**  Nvidia A100 GPU with 40 GB of memory. Similar to the Perlmutter (113 Pflop/s) [84], Leonardo (304 Pflop/s) [85] and JUWELS (70 Pflop/s) [86] systems.

**H100**  Nvidia H100 GPU with 80 GB of memory. This is a recent release by Nvidia and a likely target for next generation supercomputers.

**MI100**  AMD MI100 GPU with 32 GB of memory.

**1/2×MI250**  AMD MI250 GPU with 32 GB of memory. Similar to the Frontier (1.6 Eflop/s) [87], LUMI (531 Pflop/s) [88] and Adastra (61 Pflop/s) [89] systems. Each GPU has two tiles. In PEPPER, each tile acts as an independent device. Therefore we choose to utilize a single tile for our performance tests, which is why we include "1/2×" in our label.

**PVC**  Intel Data Center GPU Max Series with 128 GB of memory. This is part of the Sunspot testbed [90] of the Aurora (1.0 Eflop/s) systems [91]. Sunspot is a pre-production supercomputer with early versions of the Aurora software development kit.

Using a portability framework like Kokkos in the PEPPER event generator is a novel feature for a production-ready parton-level event generator, and for much of the HEP software stack in general. We have therefore tested the performance of the Kokkos PEPPER variant against both a native CUDA and a native single-threaded CPU implementation, and looked at results on an A100 GPU and an Intel Core i3-8300 CPU at 3.70 GHz CPU. When comparing the event throughput of the Kokkos and native CUDA implementations on the A100 GPU, we find that the performance of the Kokkos variant agrees within a factor of two, with the performance gap disappearing as computational complexity increases, i.e. with increasing process multiplicity. Generally, the native algorithm outperforms the Kokkos version. This result establishes that using a portability framework is possible without compromising significantly on performance when using a GPU, while at the same time giving access to a much wider range of architectures and computing paradigms. With that, we only show Kokkos result in the following. We note, however, that our current Kokkos implementation on the CPU does not utilize vectorization capabilities, while our native C++ implementation does so with the help of the VCL library [92], which provides significant speedups on the CPU. Because the focus of the analysis here is on portability, these improvements are not yet included in Figs. 4 and 5. We expect to provide a Kokkos implementation with vectorization capabilities in the future. For a more in-depth discussion of CPU vectorization of PEPPER, see App. E.

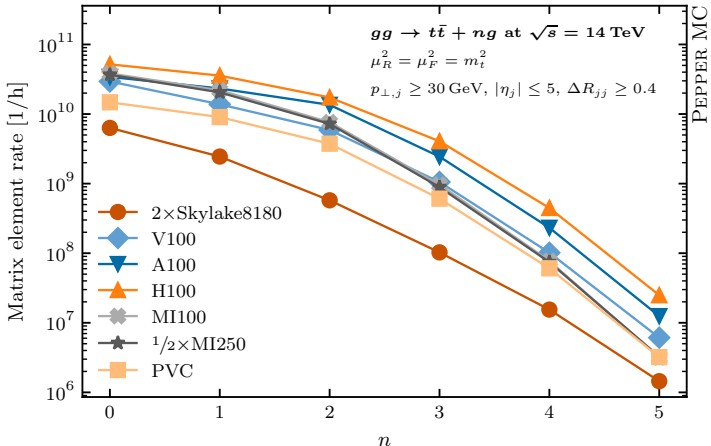

Figure 4: Comparison of the throughput achieved for the $gg \to t\bar{t} + ng$ process for the matrix element evaluation on different computing architectures. For details, see the main text.

First we study the performance on different architectures for the evaluation of weighted parton-level $gg \to t\bar{t} + ng$ events for $n = 0, \ldots, 5$ at a fixed centre-of-mass energy, thus testing the matrix element throughput of PEPPER in a wide range of parton multiplicity. The results are shown in Fig. 4. For each of the tests, we run an initial set of test runs, probing the ideal number of events that are processed simultaneously. We find that PEPPER achieves at least an order of magnitude higher throughput on the H100 GPU than on two 28-core Skylake CPUs. The results for the other accelerators lie between the H100 and the 2×Skylake8180 result. The performance on MI100/250 scales slightly worse than the Nvidia GPU with the number of additional gluons and thus with the memory footprint of the algorithm, but overall the scaling with multiplicity is qualitatively similar on all architectures. The results show that PEPPER's matrix-element evaluation is successfully ported to a wide range of hardware from different vendors.

Our next test addresses the generation of full parton-level events including unweighting, event write-out and partonic fluxes in $pp \to e^+ e^- + n$ jets and $pp \to t\bar{t} + n$ jets. The event rates are presented in Fig. 5. The numeric results are tabulated for reference in App. A. Overall, the results are similar to the ones presented in Fig. 4. One difference is that the event rates are more on par for very low multiplicities $n = 0, 1$. This is because the output of the events becomes the dominant part of the simulation due to the high phase-space efficiency and the very large matrix-element throughput on GPUs. Details are discussed in App. B. Ignoring the lowest two multiplicities, we find that the H100 event rates are 20 to 50 times higher than the 2×Skylake8180 ones. Another difference to the matrix element only case is that the scaling with multiplicity is steeper due to the quickly decreasing unweighting efficiencies. Again, the scaling is similar on all architectures. These results show that the entire PEPPER pipeline of parton-level event generation and writeout is successfully ported to a wide range of hardware.

## 5.3 Scaling to many nodes

Finally, we perform a weak scaling test of Pepper on the Polaris system at ALCF [93]. Polaris is a testbed to prepare applications and workloads for science in the exascale era and consists of 560 nodes with one AMD EPYC Milan processor and four Nvidia A100 GPUs each. The nodes have unified memory architecture and two fabric endpoints, the system interconnect

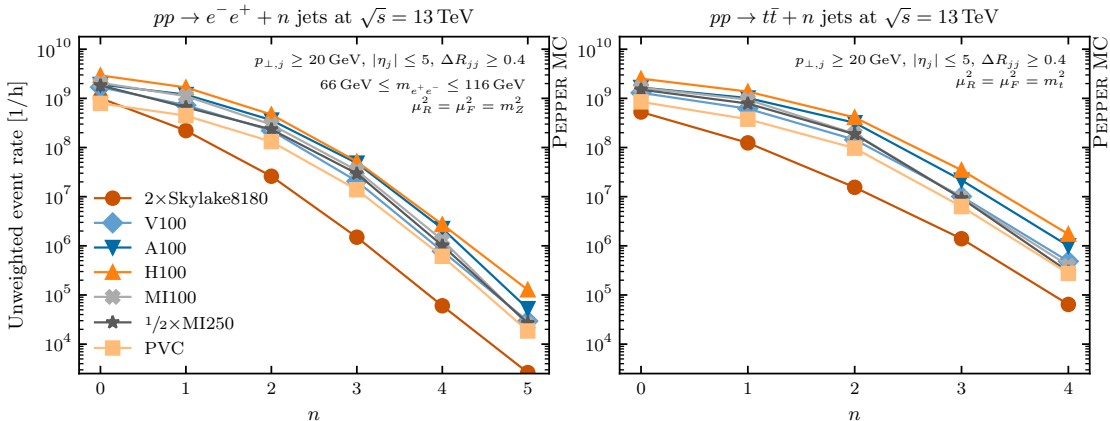

Figure 5: Comparison of the unweighted event generation and storage rates achieved for the $pp \to e^+e^- + n$ jets (left) and $pp \to t\bar{t} + n$ jets (right) processes on different computing architectures. For details, see the main text.

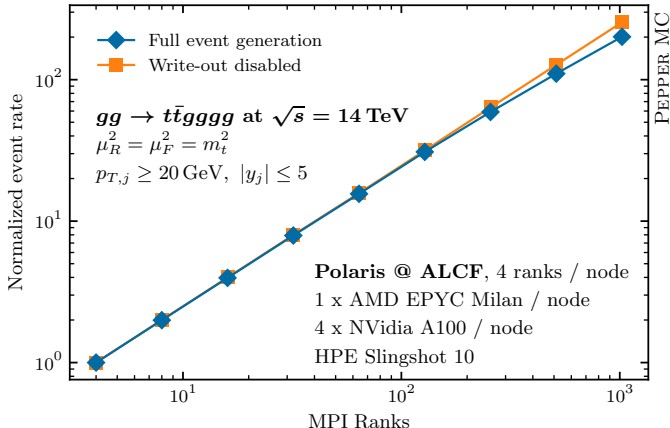

Figure 6: Scaling test of PEPPER event generation on the Polaris system [93] at ALCF. The event rates for the $gg \to t\bar{t}gggg$ process are shown, normalized to the rate on 4 MPI ranks. See the main text for details.

is a HPE Slingshot 10 network at the time of our study. For our test, we hold the number of generated events per node constant and increase the number of nodes. We measure the number of unweighted events in $gg \to t\bar{t}gggg$ production. The gluon flux is evaluated using the LHAPDF library with its builtin MPI support enabled. Figure 6 shows the event rate as a function of the number of MPI ranks, normalized to the rate on a single node. Here the number of MPI ranks is equal to the number of GPUs. We notice that the scaling is violated starting from about 512 ranks. However, we observe that the scaling violation is entirely due to the event output via the HDF5 library. We expect to be able to alleviate this problem in the future through performance tuning of HDF5, similar to the effort reported in [40].

## 6  Summary and Outlook

PEPPER is a comprehensive framework for production-level parton-level event generation for collider physics on various computing architectures. It includes matrix-element calculation, phase-space integration, PDF evaluation, and processing of events for storage in event files. In addition to its core functionalities, in [40], an extension of existing file formats was introduced and demonstrated to generate merged samples with Pepper.

Next-to-Leading Order calculations are of great importance to the LHC. By using a generalized unitarity method, one can utilize PEPPER, with minor extensions, to calculate the coefficients of the master integrals and the rational part of the one-loop matrix elements. The numerical aspect of this method, which is well-suited for GPU implementation, involves solving sets of linear equations generated by evaluating leading-order matrix elements with possible complex external momenta in integer higher dimensions.

Another important note, is that these calculations use double floating-point precision to help ensure sufficient accuracy for momentum conservation, and the higher order corrections require similar precision or sometimes even quad floating-point precision to ensure numerical stability [94–96]. However, AI applications are dictating the direction of future heterogeneous computing systems, which tend to require either half or single floating-point precision [97]. To address these concerns we can leverage techniques developed for efficiently using single (double) floating-point precision to obtain double (quad) floating-point precision [98–100], or develop other custom data types to handle these calculations, as done in the lattice QCD community [101].

The development outlined in this paper marks a major milestone identified by the generator working group of the HEP Software Foundation [4, 5] and by the HEP Center for Computational Excellence [102]. It frees up valuable CPU resources for the analysis of experimental data at the Large Hadron Collider. The impact of parton-level event generation to the projected shortfall in computing resources during Run 4 and 5 in the high-luminosity phase of the LHC is significantly alleviated. In addition, we enable the Large Hadron Collider experiments to utilize most available (pre-)exascale computing facilities, which is an important step towards a sustainable computing model for the future of collider phenomenology.

The PEPPER event generator is ready for production-level use in many processes to be simulated at high fidelity at the LHC, i.e. $\ell^+\ell^-$+jets, $t\bar{t}$+jets, pure jets and multiple heavy quark production (e.g. $b\bar{b}b\bar{b}$+jets or $t\bar{t}b\bar{b}$+jets). The combination with the particle-level event generation framework in [20] makes it possible to process events with PYTHIA 8 [70] or SHERPA 2 [69]. An extension of PEPPER to processes such as $\ell\bar{\nu}_\ell$+jets, $VV$+jets, $\gamma\gamma$+jets and to other reactions with high computing demands will become available in the short term. The compute performance of the CPU version of PEPPER is better than that of COMIX, one of the leading and most widely used automated matrix element generators for the LHC. We have shown, for a variety of architectures, that PEPPER can efficiently generate parton-level events,

and we have demonstrated scalability up to 512 Nvidia A100 GPUs on the Polaris system at ALCF.

## Acknowledgments

This research was supported by the Fermi National Accelerator Laboratory (Fermilab), a U.S. Department of Energy, Office of Science, HEP User Facility. Fermilab is managed by Fermi Research Alliance, LLC (FRA), acting under Contract No. DE–AC02–07CH11359. The work of M.K. and J.I. was supported by the U.S. Department of Energy, Office of Science, Office of Advanced Scientific Computing Research, Scientific Discovery through Advanced Computing (SciDAC-5) program, grant "NeuCol". This research used resources of the Argonne Leadership Computing Facility, which is a DOE Office of Science User Facility under Contract DE-AC02-06CH11357. The work of T.C. and S.H. was supported by the DOE HEP Center for Computational Excellence. E.B. and M.K. acknowledge support from BMBF (contract 05H21MGCAB). Their research is funded by the Deutsche Forschungsgemeinschaft (DFG, German Research Foundation) – 456104544; 510810461. This work used computing resources of the Emmy HPC system provided by The North-German Supercomputing Alliance (HLRN). M.K. wishes to thank the Fermilab Theory Division for hospitality during the final stages of this project. This research used the Fermilab Wilson Institutional Cluster and computing resources provided by the Joint Laboratory for System Evaluation (JLSE) at Argonne National Laboratory. We are grateful to James Simone for his support.

## A    Tabulated performance results

Table 1 lists the baseline performance data shown graphically in Fig. 3. While the figure shows ratios, we list here the absolute event rates found for COMIX and PEPPER using single-threaded execution. For additional details, see Sec. 5.1.

Table 2 lists the unweighted events rates on different computing architectures. This is the raw data for Fig. 5. For additional details, see Sec. 5.2.

## B    Runtime distribution

In Sec. 5.2, we found that the event rates for the lowest multiplicities are comparably similar across the different computing architectures, see Fig. 5. To understand this better, we plot the fractions of computing time spent in different components of the event generation in Fig. 7. The different components studied are the Berends–Giele recursion for the matrix element evaluation, the event output, the evaluation of the strong coupling and partonic fluxes, and the phase space generation. On the left side we plot the data of the 2×Skylake8180 architecture, while on the right side we plot the data of the Nvidia H100 architecture. These are representative of a (two-chip) CPU system and a GPU architecture. While on the CPU the majority of time is always spent on the matrix element evaluation, we have a different situation on the GPU. Here, for the lowest three multiplicities the majority of time is spent on the event output. This is because the matrix element evaluation rate on the GPU is so high that the overall rate is now constrained by the event file write-out. However, in the current implementation of GPU write-out, only a single CPU core is used. Therefore, the write-out rate could be improved by utilizing the idle CPUs on each node, an implementation of this procedure is left to a future version of PEPPER.

| Events / hour | COMIX* | | PEPPER |
| --- | --- | --- | --- |
| | COMIX | CHILI | |
| $pp \to e^+e^- + 1j$ | 1.2e7 | 9.0e6 | 7.2e7 |
| $pp \to e^+e^- + 2j$ | 1.2e6 | 1.0e6 | 7.2e6 |
| $pp \to e^+e^- + 3j$ | 1.1e5 | 7.5e4 | 5.0e5 |
| $pp \to e^+e^- + 4j$ | 6.2e3 | 4.0e3 | 1.4e4 |
| $pp \to e^+e^- + 5j$ | 3.8e2 | 2.0e2 | 3.0e2 |

| Events / hour | COMIX* | | PEPPER |
| --- | --- | --- | --- |
| | COMIX | CHILI | |
| $pp \to t\bar{t} + 0j$ | 9.0e6 | 1.2e7 | 3.6e7 |
| $pp \to t\bar{t} + 1j$ | 2.4e6 | 3.3e6 | 1.2e7 |
| $pp \to t\bar{t} + 2j$ | 2.1e5 | 3.8e5 | 2.2e6 |
| $pp \to t\bar{t} + 3j$ | 1.2e4 | 2.9e4 | 1.3e5 |
| $pp \to t\bar{t} + 4j$ | 8.1e2 | 1.9e3 | 6.4e3 |

Table 1: Comparison of the event rates achieved for $pp \to e^+e^- + n$ jets (left) and $pp \to t\bar{t} + n$ jets (right) by COMIX, combined with its default and with the basic CHILI phase-space generator, and PEPPER. The asterisk in the COMIX label indicates that it is run in a non-default but more efficient mode, splitting the process sum into different Monte-Carlo channels, grouping by gluons, valence and sea quarks. The results were generated on a single core of an Intel Xeon E5-2650 v2 CPU. For details, see Sec. 5.1.

| Events / hour | 2×Skylake8180 | V100 | A100 | H100 | MI100 | 1/2×MI250 | PVC |
| --- | --- | --- | --- | --- | --- | --- | --- |
| $pp \to t\bar{t} + 0j$ | 5.3e8 | 1.3e9 | 1.7e9 | 2.5e9 | 1.6e9 | 1.5e9 | 8.5e8 |
| $pp \to t\bar{t} + 1j$ | 1.2e8 | 6.2e8 | 1.0e9 | 1.4e9 | 9.4e8 | 7.8e8 | 3.8e8 |
| $pp \to t\bar{t} + 2j$ | 1.6e7 | 1.4e8 | 3.2e8 | 4.1e8 | 1.9e8 | 1.9e8 | 9.7e7 |
| $pp \to t\bar{t} + 3j$ | 1.4e6 | 1.0e7 | 2.2e7 | 3.5e7 | 9.4e6 | 9.2e6 | 6.3e6 |
| $pp \to t\bar{t} + 4j$ | 6.4e4 | 4.8e5 | 1.0e6 | 1.7e6 | 4.0e5 | 3.0e5 | 2.8e5 |
| $pp \to e^-e^+ + 0j$ | 1.0e9 | 1.7e9 | 1.9e9 | 2.9e9 | 2.1e9 | 1.8e9 | 7.9e8 |
| $pp \to e^-e^+ + 1j$ | 2.2e8 | 7.3e8 | 1.2e9 | 1.7e9 | 1.1e9 | 6.6e8 | 4.5e8 |
| $pp \to e^-e^+ + 2j$ | 2.6e7 | 2.2e8 | 3.6e8 | 4.7e8 | 2.9e8 | 2.3e8 | 1.3e8 |
| $pp \to e^-e^+ + 3j$ | 1.5e6 | 2.1e7 | 4.8e7 | 5.2e7 | 3.4e7 | 3.0e7 | 1.4e7 |
| $pp \to e^-e^+ + 4j$ | 6.0e4 | 7.8e5 | 2.2e6 | 2.7e6 | 1.3e6 | 1.0e6 | 6.1e5 |
| $pp \to e^-e^+ + 5j$ | 2.6e3 | 2.9e4 | 5.3e4 | 1.3e5 | 2.5e4 | 2.7e4 | 1.8e4 |

Table 2: Comparison of the unweighted event generation (and storage) rates achieved for the $pp \to t\bar{t} + n$ jets and $pp \to e^+e^- + n$ jets processes on different architectures.

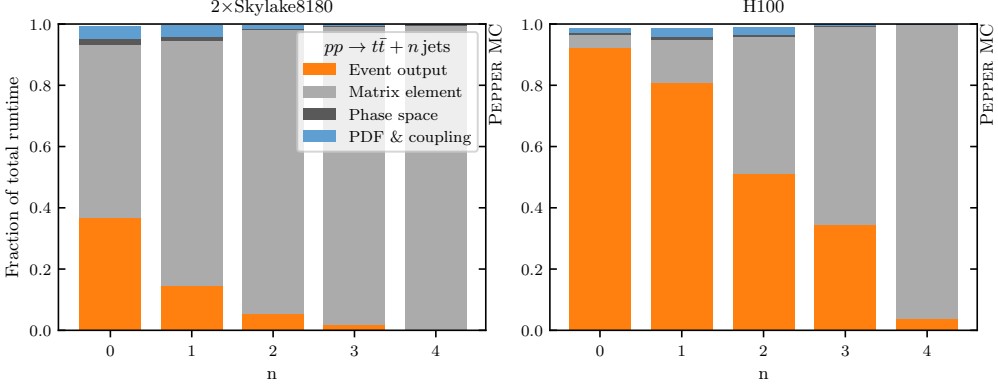

Figure 7: The fraction of event generation runtime spent on different components of the pipeline, for the $pp \to t\bar{t} + n$ jets process. On the left, we show the data for the 2×Skylake8180 CPU, while on the right we show it for the Nvidia H100 GPU. The components are Event Output, Matrix element evaluation, Phase space sampling, and PDF and strong coupling evaluation.

## C   Parton-level validation

We repeat the validation of Sec. 4, using the same setups and observables as described there. However, here we omit the particle-level simulation steps, i.e. we only calculate parton-level events without any further processing. This compares directly the phase-space sampling and matrix-element generation implementations of PEPPER and AMEGIC. Given a correct implementation, they should give statistically compatible results.

The comparison results for the two Z+jets observables are shown in Fig. 8. For details on the presentation and the observables, we refer to Sec. 4. The agreement is again quantified using a Kolmogorov–Smirnov test. The $p$-values for each jet multiplicity can be found on the plot. They are all greater than our chosen confidence level of 5 %, i.e. we do not reject the null hypothesis of the two underlying distributions being identical.

Finally, the comparison results for the two $t\bar{t}$+jets observables are shown in Fig. 9. Again, the results of a Kolmogorov–Smirnov test are quoted on the plot, with all $p$-values being greater than 0.05. Thus, also in this case, we do not reject the null hypothesis.

## D   Data layouts and CPU performance

We study the performance of our struct-of-array (SoA) implementation when running on a single CPU thread, and compare it with an array-of-structs (AoS) one. With AoS, each event is represented by a data struct, and all events of a batch are stored in an array of such structs. Some data members of each event are themselves arrays, such as the four-momenta, or the complex currents used to matrix elements. This is opposed to an SoA implementation, where each event property is stored as an array over all events in the batch.

The two approaches yield different caching opportunities.[7] In Fig. 10, we compare their performance on an Apple M2 Pro chip for different event batch sizes for the $pp \rightarrow t\bar{t} + 3$ jets process. The performance is measured in events per hour, normalized to the SoA event rate for a batch size of one. We find small performance improvements of about 10 % when increasing the batch size to 10. . . 100 events, with the SoA performance being within a few percent with the AoS performance, indicating that good cache efficiency is achieved for both data layouts.

For a batch size of $10^3$, the performance degrades significantly for the AoS layout, which might indicate that not all events fit anymore in one of the caches. This happens earlier compared to the SoA layout, which shows a degraded performance only at a batch size of $10^4$, since in that case the CPU is able to cache only the locally relevant data, without including data for properties that are not being read or written to by the given algorithm.

Note that an AoS layout would likely perform significantly better if the iteration over the events of a given batch would take place in the outer-most loop, while in our implementation this loop is performed instead at the algorithm level. However, to study this we would need to completely restructure the code, which is beyond the scope of this study. In addition, catering for both approaches in a single codebase would likely require severe compromises when it comes to the readability and maintainability of the code.

Given the kernel-based structure of our code, we have shown that an SoA layout is preferred even for single-threaded execution on a CPU, and that it is favourable to use batch sizes of at

---

[7]You might also expect them to yield different degrees of CPU auto-vectorization. However, to achieve good performance on the GPU we had to fuse most kernels of the matrix element recursion into a larger recursion kernel, to avoid the overhead of starting a large number of very small kernels on the GPU. This means that the loop over events is a few layers removed from the inner-most loop in this case, which is of course also the most compute-intense part of the code. This seems to prevent auto-vectorization almost entirely. We study two ways to CPU-vectorize the code explicitly in App. E. None of these play a role in the current discussion though, as we have disabled the explicit vectorization for all results obtained in App. D.

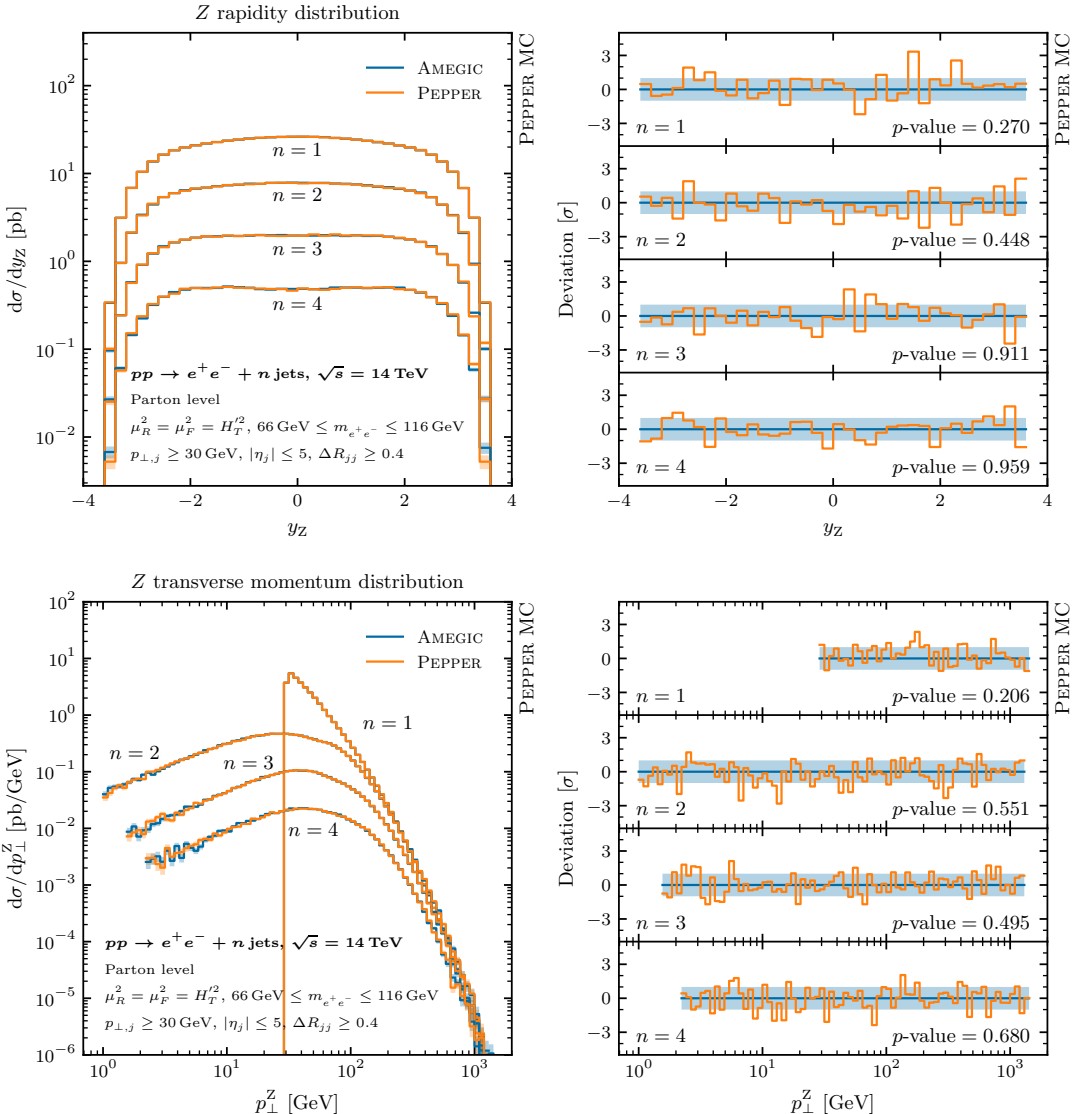

Figure 8: Parton-level validation for two observables for the $pp \to Z + n$ jets process, comparing AMEGIC results with PEPPER results. The left plots show the distributions, while the right plots show the deviations between AMEGIC and PEPPER individually for each $n$, normalized to the $1\sigma$ standard deviation of the AMEGIC result. For each $n$, the $p$-value of a Kolmogorov–Smirnov test is shown for the hypothesis that the deviations follow a standard normal distribution.

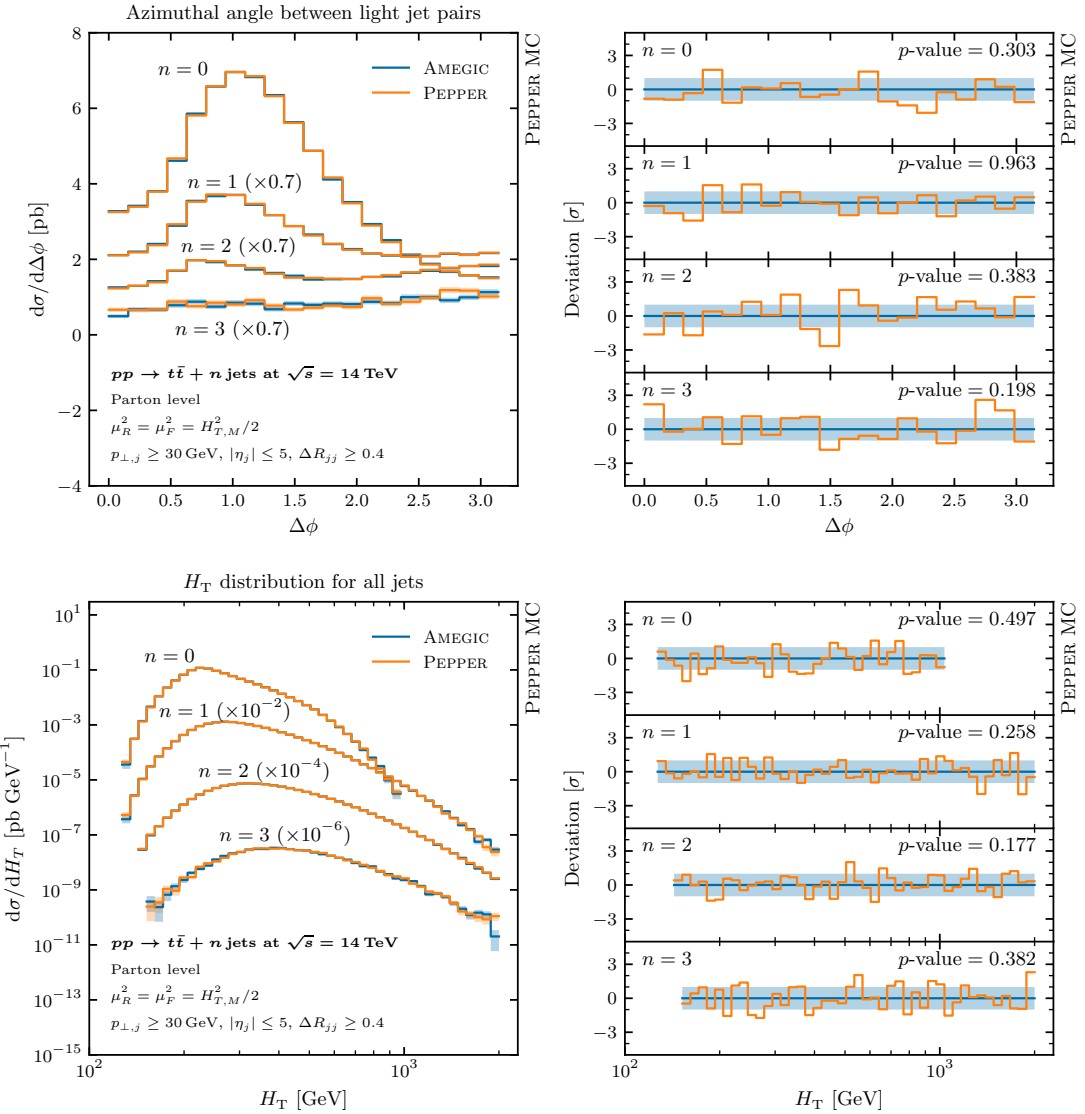

Figure 9: Parton-level validation for two observables of the $pp \rightarrow t\bar{t} + n$ jets process, comparing AMEGIC results with PEPPER results. The left plots show the distributions, while the right plots show the deviations between AMEGIC and PEPPER individually for each $n$, normalized to the $1\sigma$ standard deviation of the AMEGIC result. For each $n$, the $p$-value of a Kolmogorov–Smirnov test is shown for the hypothesis that the deviations follow a standard normal distribution.

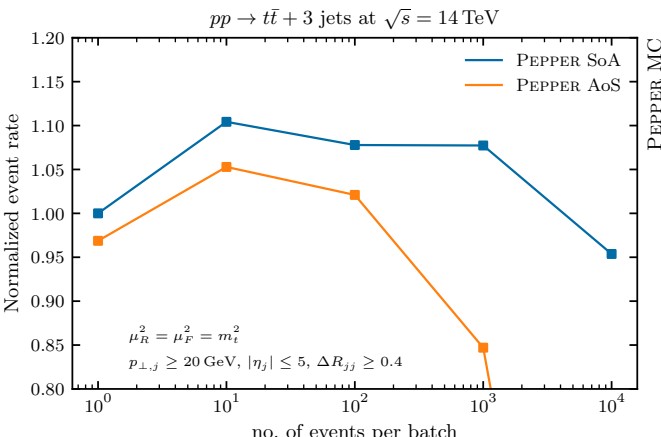

Figure 10: Scaling test of the event generation performance of PEPPER for serial CPU execution on an Apple M2 Pro chip, comparing a struct-of-array (SoA) with an array-of-structs (AoS) event data layout. The $x$ axis gives the number of events per event batch, while the $y$ axis shows the performance given in event throughput, normalised to the event throughput of the SoA layout with a batch size of one.

least ten events (and likely even more for simpler processes or larger cache sizes) to fully profit from the CPU caches.

## E CPU vectorization

As mentioned in Sec. 5.2, the native C++ implementation of PEPPER does use some explicit CPU vectorization for certain calculations. Currently, it does so by implementing real and complex four momentum classes with the help of the Vector Class Library (VCL) [92], such that all the arithmetic operations involving four momenta are compiled using vector intrinsics of the CPU (if supported by VCL). The advantage is that this is a drop-in replacement: One can simply replace the four momentum classes, and any code using these classes immediately profit from the acceleration, without any change. However, there are two disadvantages. One is that the maximum degree of parallelization is limited by the size of the objects. E.g. for AVX-512, where CPU vectors are 512 bits wide, only operations including complex four momenta (which consist of 8 64-bit numbers) would fully utilize the hardware. The second disadvantage is that only simple operations, i.e. addition, subtraction and multiplication by a scalar, are fast. Other operations such as scalar products, that need to take a sum over all components, are only expected to achieve medium efficiency [92]. We can test the performance by measuring the evaluation time of a compute kernel with non-trivial arithmetics and a sizable contribution to the overall runtime. A perfect example is the three gluon vertex kernel, which in particular calculates the complex four-component current

$$j_c = (j_a \cdot (2p_b + p_a)) \cdot j_b + (j_a \cdot j_b)(p_a - p_b) - (j_b \cdot (2p_a + p_b)) \cdot j_a. \tag{E.1}$$

Here, the $j$ are complex four-component currents and the $p$ are real four momenta, and the subscripts $a, b, c$ label the three outgoing particles of the vertex. This mixes real and complex vectors and simple component-wise and component-mixing operations. We choose to measure the evaluation time in $pp \rightarrow t\bar{t}jjjj$ production on an Intel Xeon Gold 6430 with its support for AVX-512 intrinsics. For 9600 weighted events, the evaluation of the three gluon vertexes

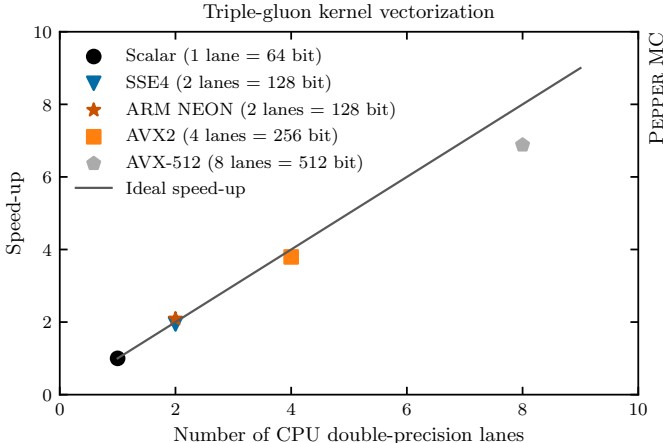

Figure 11: CPU vectorization speed-ups of the three gluon vertex evaluation for different vector intrinsics, using a preliminary version of PEPPER that vectorizes calculations across events using the Highway library [103]. The speed-ups are normalized to the time needed to evaluate the vertex without CPU vectorization, i.e. using "scalar" code. The SSE4, AVX2 and AVX-512 intrinsics have been tested on an Intel Xeon Gold 6430 chip, while an Apple M2 Pro chip has been used to test the ARM NEON intrinsics. The evaluation times have been measured during the generation of $pp \to t\bar{t}jjjj$ events. Note that the CPU vectorization in the released versions of PEPPER is less pronounced, as explained in the main text.

requires 4.8 (15.1) seconds with (without) VCL accelerated four momentum classes enabled. Thus, a speed-up of about a factor of three is observed, which falls significantly short of the theoretical factor of eight for double precision arithmetics with 512 bit wide CPU vectorization.

The implemented method discussed so far vectorizes calculations for each individual event. Given the SoA data layout of PEPPER, see App. D, it is straightforward to study a different way to vectorize the code, namely to vectorize calculations across several events. This eliminates both disadvantages discussed above: It scales arbitrarily with the size of the CPU vectors, as long as we set the event batch size to some integer multiple of the CPU vector size (measured in double-precision lanes, i.e. multiples of 64 bit). Also, all operations are fully parallel, as the calculations for the events are completely independent from one another. It can therefore be expected that more significant speed-ups can be achieved. Ideally, if the overhead is negligible, the speed-up is equal to the number of CPU double-precision lanes. As a preliminary test, we port the evaluation of the three gluon vertex to use vector intrinsics via the Highway library [103], which supports intrinsics on most common platforms.

We again measure the time for the evaluation of the three gluon vertices in $pp \to t\bar{t}jjjj$ production, on the same Intel Xeon Gold 6430 machine, which supports AVX-512 intrinsics (8 lanes), but also AVX2 intrinsics (4 lanes) and SSE4 intrinsics (2 lanes), such that we can study the scaling with the number of lanes. In addition, we measure it for an Apple M2 Pro chip with ARM NEON intrinsics (2 lanes), to establish the portability advantage of the Highway library as compared to VCL, which does not support ARM and does not plan to do so [92]. To calculate a speed-up, we divide by the time needed on the respective chip when disabling Highway's use of intrinsics (the operators are then defined by standard C++ code). Note that each time measurement is repeated three times, and then the average is used. We only find small variations across the repetitions which are irrelevant for the present discussion.

Figure 11 shows the achieved speed-ups versus the number of lanes for the three gluon

vertex evaluation time for the various vector intrinsics used. While the speed-up is nearly ideal in the 2-lane case, we find speed-ups of 3.8 and 6.9 for the 4- and 8-lane cases, respectively, indicating that overheads become a relevant factor, possibly exacerbated by a reduction of the clock-speed in the AVX-512 case [104]. However, we still get close to the ideal speed-up in all tests and therefore deem this a successful demonstration that PEPPER can be vectorized in this manner, with only minor modifications that can be implemented on a kernel-to-kernel basis. The only drawback so far is that in our initial implementation for this demonstration, we lose readability and maintainability of the kernel code because we decompose the calculation of the four currents into its 4 real and 4 imaginary components instead of relying on operators that work directly with the complex currents and four momenta, thus losing the encapsulation of the details of the calculation. We leave it to future work and applications to determine if this is a compromise worth making, or to find a way to wrap the operations in such a way as to sufficiently retain the expressiveness of the kernel code, while still achieving comparable speed-ups.

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
