# Peer review of "A Portable Parton-Level Event Generator for the High-Luminosity LHC"

_SciPost Physics_

## Round 2 · Referee Report · Anonymous (Referee 1) · 2024-1-19

Strengths

  • The paper is very timely given the current HPC facility and given HL-LHC need.
  • Implementation is very strong both in terms of algorithm and in terms of actual code portability.
  • comparison of the code made on numerous hardware and process of different complexity.

Weaknesses

  1. The paper is over-selling some points (see full report below for the ones I detected).
  2. The paper does not recognize correctly the previous (complete) implementation on GPU (some of them are 10 years old).
  3. Physics validation need also to be done at parton-level (not particle level --at least not only--)
  4. They are no proof that GPU are used efficiently (no roof-plot or any of the standard plot/...)
  5. This work is only LO.

Report

The work done by the authors is impressive and certainly deserve publication and recognition. The content of the paper fits perfectly the acceptance criteria.

However, the authors seem to be afraid that their work does not get enough recognition and are over-selling their work both in terms of innovation (one should properly recognize the first working GPU implementation done 10 years ago) and in terms of impact (the contrast between the depicted situation in the introduction and the one in the conclusion is over-exaggerated -- and none are true--). This type of exaggeration should not be accepted for publication.

In terms of physics validation, the authors made a curious choice to present validation at particle level. While this is great to convey the message that the code is ready for LHC production. It does introduce additional noise/source of statistical uncertainty for the comparison. While this is clearly a minor issue, this is also something that is easy for the authors to do (if they have not done it already).

In terms of hardware validation, the authors do prove that they have a good MPI scaling but do not prove that they have a good scaling on GPU. They state, on page 7, that they are memory bound, but I doubt that this is true for all the processes tested. Also, the statement about improvement with single thread with SOA is in itself surprising (and surprisingly large) and the authors should comment if this speed-up come from vectorization, caching on RAM (and then comment on the amount of RAM used) or something else. While such technical details might not be important for most physicist, they need to be documented in the publication.

Given the point above, my recommendation is therefore to ask for a minor revision of the paper, given that all my requested change should be quite easy/straightforward to handle.

Requested changes

  1. The authors should tune abstract, introduction and conclusion in order to
  2. better indicate the existence of previous work. In particular, I do not understand why they claim to be the "first" while they quote the previous work of [25] which is the first complete implementation on GPU as far as I know. They should also cite the much more recent MadFlow implementation.
  3. weaken the statement that event simulation is the limiting factor in the introduction.
  4. weaken the statement, in the conclusion, that parton-level generation will no longer contribute. (NLO and NNLO will still be problematic, and many LO computation are performed for BSM physics which is not supported by pepper)
  5. comment on the fact that state of the art are today NLO and not LO.

  6. Present some validation at parton-level.

  7. Give more technical number on the usage of the GPU (like batch size/rate of divergence of thread/occupancy/…) and how much of the peak performance (and bandwidth) are used for each of the GPU.

Additionally, I would like to suggest the authors to add some points to improve the clarity of the paper on the following points:

  1. Add some comment on dynamical scale choice (in particular in view of CKKW type of merging, which is surprisingly not mentioned at all in the paper).

  2. explain why SOA is helping in case of single thread while AOS should be a better match (but if vectorization is applied).

  3. The workflow corresponding to when cuts are evaluated/unweighting performed/... is not well explained in the paper. One point which is not clear is that you seem to evaluate the matrix-element even if the phase-space points do not pass the cuts. If this is True, it would be important to mention how much of the GPU time is wasted due to that.

  4. Related to the previous point, the correlation within a block of 32 events seems to indicate that you write all event and not only unweighted events (which the authors state later that this is not the case)... I guess that the re-ordering is actually only needed for low multiplicity.

  5. Comment on the floating point precision need of your GPU (especially since you are motivating the deployment of GPU by AI software which likes to use half precision).

---

## Round 2 · Referee Report · Anonymous (Referee 2) · 2024-2-12

Strengths

  1. The paper presents a portable leading-order event generator framework capable of utilising a number of computing architectures.

  2. The event generator is benchmarked against the widely used Sherpa+Comix framework, and outperforms it in almost all the cases considered in the paper.

  3. The framework seems very promising in terms of scalability and performance on modern GPU architectures.

  4. This development is very timely, given the the prevalent discussions taking place within the community currently.

  5. The code is publically available on github.

Weaknesses

  1. The physics novelty of the paper is rather limited, relying on previously known results for both phase space and matrix element generation.

  2. In the paper a number of different architectures are compared. This clearly shows the portability feature of the framework, but it is not clear that performance can be assessed in this way, as the architectures are not equivalent.

  3. The framework can only handle leading-order generation. Some discussion of the possibility/difficulties of going to next-to-leading order would have been nice, given that this is typically the precision used by the experiments.

  4. It would have been interesting to see a discussion of the environmental benefit of the approach, given that this was raised in the introduction.

  5. Although the code is publically available, the documentation is not complete enough that a user can readily get started with the framework.

Report

The paper documents a new framework, Pepper, for leading-order event generation. The main novelty of the framework is that it is highly portable, and hence that it can take advantage of modern High Performance architectures (e.g. GPUs). This is a much needed development, and an essential part of reducing the computational impact of Monte Carlo generators, both in terms of resources and their environmental impact.

The paper is clearly written and as such the paper deserves to be published in SciPost. However, I have a few questions/comments that should be addressed first.

Requested changes

  1. The authors mention environmental impact in the introduction. Although a detailed assessment of the impact if outside the scope of the paper, it would be good to see some discussion of this in relation to the portability .For instance in terms of the power consumption of the various architectures and the time spent on equivalent computations.

  2. Related to that, in section 5.2 the authors compare the performance of Pepper on a number of different architectures. Although this nicely confirms the portability aspects of the code,i t is not clear what one can conclude in terms of performance, since this is not an apples to apples comparison. Could the authors try to address that?

  3. It makes sense to limit the implementation to leading-order for now, but given that almost all analyses at the LHC use next-to-leading order (or beyond) the authors must at least discuss the possibilities of extending their framework in that direction, and outline what the difficulties might be. It would also be interesting to understand if merged samples could be generated with Pepper.

  4. This one doesn't have to be addressed, but I just wanted to note that it isn't clear that one needs to be on the "native" branch (or download the native release) in order to compile without Kokkos. I managed in the end, although the code did not compile with my Cuda installation (Ubuntu + nvcc 12.3). After compiling I also could not find any documentation on how to run the code. I guess this will be added at a later stage?

---

## Round 3 · Referee Report · Anonymous (Referee 1) · 2024-6-11

Report
Thanks to the authors for updating their document taking into account both comments. This impressive paper should therefore be mostly ready for publication.
However, I would like to ask further clarification on a couple of points from my previous report.
- Concerning the new Appendix D and Figure 10: As the author knows pretty well, a good usage of the hardware (M2 chip here) and a good memory layout, should lead to a factor of two speed up (speed-up of eight for the most modern HPC hardware). The authors do not observe such factor (either due to the memory layout, due to the code algorithm or to Kokkos itself or ...).
This weakens the "ecology" point of the paper, since it shows not fully efficient use of the hardware (but still better or on part to current used code). Additionally, if the issue is related to non-optimal memory layout, this should also impact GPU performance.
While fixing the issue might be too complicated for this paper, I think that that author should comment on this, on the probable cause and if they plan to fix it in the future.
-
Given the previous point, I want to reiterate, my previous request to see hardware reports showing how efficiently the hardware is used. While I understand that the authors do not want to do a technical paper, such information is crucial for the claim that portability is helping the environment. Those numbers can be provided as supplementary material online and might not even need to be formatted with text/... However, I will not block the publication if this is not provided.
-
The authors gave convincing argument about their strategy for their handling of the cuts but only as a reply to the referee. I think that the paper should include such arguments.
Requested changes
-
Comment on the fact that the code use poorly CPU+RAM paradigm (very minor)
-
Comment on why Appendix D seems to not use the CPU correctly and what are the reason/plan to fix it (or why you consider this reasonable).
Recommendation
Ask for minor revision

---

## Round 3 · Author Response

We thank the two referees very much for their detailed reports and valuable suggestions. We believe that we have addressed all points which have been raised in our new version of the draft (v3). In the following we will give our answers to each of the requested changes, which we repeat here for easier reference. The citations numbering conforms to the references given in the resubmitted draft (v3).
Referee report 1 - requested changes and our replies:
The authors should tune abstract, introduction and conclusion in order to
better indicate the existence of previous work. In particular, I do not understand why they claim to be the ”first” while they quote the previous work of [25] which is the first complete implementation on GPU as far as I know. They should also cite the much more recent MadFlow implementation.
Our reply: We removed “first” in the abstract and the conclusions, and the MadFlow references are now added in the introduction.
weaken the statement that event simulation is the limiting factor in the introduction.
Our reply: We do not claim that event generation is the limiting factor, only that it “can become a limiting factor”. This statement is in agreement with the assessment by experimental collaborations at various recent workshops and planning exercises, see for example the ATLAS and CMS presentations at https://indico.cern.ch/event/1312061/, or refs. [3-5] (we have added [3] in the text). We also mention now the developments of modern approaches to detector simulation and reconstruction, which is expected to reduce the relative cost of these steps in the simulation pipeline (and thus increase the relative cost of event generation), and cite these approaches.
weaken the statement, in the conclusion, that parton-level generation will no longer contribute. (NLO and NNLO will still be problematic, and many LO computation are performed for BSM physics which is not supported by pepper)
Our reply: We have weakened the statement in the conclusions. Please note, however, that the authors have evaluated this problem very carefully in [13], and that the simulation in realistic LHC setups is dominated by the high-multiplicity tree-level components, not by NLO calculations.
comment on the fact that state of the art are today NLO and not LO.
Our reply: We have added a sentence in the introduction to stress that delivering a portable parton-level generator at LO is not only a natural starting point, but of immediate utility because tree-level matrix elements and phase-space sampling have the largest computational footprint even in state-of-the-art multi-jet merged calculations where the lower multiplicities are evaluated at NLO, as has been reported in [13, 39]. Further, we have acknowledged the importance of NLO calculations at the LHC in the Summary and Outlook, and discuss possibilities and challenges when extending Pepper towards NLO.
2. Present some validation at parton-level. Our reply: This is now added as a new App. C. The appendix is referred to from the validation section. The new results show that the parton-level only data is also statistically compatible, directly comparing Pepper vs. Amegic without any further down-stream simulation steps.
3. Give more technical number on the usage of the GPU (like batch size/rate of divergence of thread/occupancy/. . . ) and how much of the peak performance (and bandwidth) are used for each of the GPU. Our reply: We have performed detailed technical performance studies of the core algorithms in our previous paper on the gluon-only case [32], which we also supplemented with NVidia NSight performance benchmarks. The conclusions drawn in [32] are still valid. In particular we are still memory bound, because the introduction of fermions requires the use of complex currents and thus only intensifies the memory requirements. The focus of this work was rather on the generalization to a tool that can be used for physics applications, its portability and accompanying performance comparisons across different state-of-the-art architectures. We think that the performance presented here is excellent compared to currently available tree-level matrix element generators and significantly alleviates the phase-space and matrix-element bottlenecks discussed in [34]. Therefore, in our opinion, in-detail technical benchmarks of GPU runs—while certainly interesting—are not urgently needed, and can be done in a future study. We have added a second paragraph in the introduction of Sec. 5, to better define the scope of this work's performance studies, and refer to [32] for more technical studies.
4. Add some comment on dynamical scale choice (in particular in view of CKKW type of merging, which is surprisingly not mentioned at all in the paper). Our reply: • We have modified Sec. 3.1 to mention the possibility of dynamical scale choices. We have also noted that the strong coupling is also evaluated in parallel using the modified LHAPDF version. • The fact that the scales can by evaluated dynamically is now also listed as step 5 of the new enumeration of the parallelized event generation steps in Sec. 3.3. • We now refer to [39] in the Conclusions and Outlook, for a discussion and application for a multi-jet merged calculation. Therein, parton-level events by Pepper are read in by Sherpa, which perform the CKKW-type merging.
5. explain why SOA is helping in case of single thread while AOS should be a better match (but if vectorization is applied). Our reply: • There likely is a good degree of vectorization across events, since our event loop is around relatively simple compute kernels (and not the outer-most loop). However, we did not study this systematically. • We now realize that our comparison was not fair. Our comparison number came from comparing to an older prototype of our code. While that meant we did compare to an AOS data layout, the speed-up might come from various sources, so our statement is too strong. • However, now, given that our data access goes through getter and setter methods, we were able to quickly implement an AoS variant of Pepper. Note that this only changes the data layout, but not the hierarchy of the loops (i.e. the event loop is still close to the individual algorithms, which should favour an SOA layout). With that, we were able to generate new comparison numbers and quote those. We have put the details into App. D, and only mention the main takeaways in the main text of Sec. 3.3, in order to keep the flow of the text intact.
6. The workflow corresponding to when cuts are evaluated/unweighting performed/... is not well explained in the paper. One point which is not clear is that you seem to evaluate the matrix-element even if the phase-space points do not pass the cuts. If this is True, it would be important to mention how much of the GPU time is wasted due to that. Our reply: The event generation steps are now listed in Sec. 3.3. The question about the cuts is specifically addressed below the list of steps. While we indeed evaluate the matrix elements for events that do not pass the cuts (or, actually, simply return early for the kernels where the event weight is set to zero by the cuts, such that these threads will be idle most of the time), this is not a severe issue for the applications (processes and parton-level cuts) studied in the paper. Even for Z+5j and tt+4j with standard cuts, the phase-space efficiencies are above 85 %. We therefore conclude that a more complex strategy is not required at this point.
7. Related to the previous point, the correlation within a block of 32 events seems to indicate that you write all event and not only unweighted events (which the authors state later that this is not the case)... I guess that the re-ordering is actually only needed for low multiplicity. Our reply: This is true, we do not write out every event but only the unweighted ones. Thus, the referee is correct that correlation is indeed more prevalent for the lower multiplicities, if unweighting is enabled (which is the default), since they typically have larger unweighting efficiency and hence often unweight multiple events per block of 32. In the text, we have now explained this more clearly.
8. Comment on the floating point precision need of your GPU (especially since you are motivating the deployment of GPU by AI software which likes to use half precision). Our reply: During these calculations, the floating point precision required is typically double precision to ensure sufficient momentum conservation, and at higher orders to ensure sufficient precision on the loop corrections. We have added a short paragraph in the outlook discussing the precision requirements of our calculation, the issues that may arise from AI influences on hardware developments, and some approaches we are considering in addressing these concerns.
Referee report 2 - requested changes and our replies:
1. The authors mention environmental impact in the introduction. Although a detailed assessment of the impact if outside the scope of the paper, it would be good to see some discussion of this in relation to the portability .For instance in terms of the power consumption of the various architectures and the time spent on equivalent computations. Our reply: We have added a short discussion in the introduction about fully using HPC systems, which have a variety of architectures. Also, we highlight that providing a portable framework would enable the adoption of new more efficient hardware with minimal testing requirements.
2. Related to that, in section 5.2 the authors compare the performance of Pepper on a number of different architectures. Although this nicely confirms the portability aspects of the code, it is not clear what one can conclude in terms of performance, since this is not an apples to apples comparison. Could the authors try to address that? Our reply: In the second paragraph of Section 5.2, we described this apples-to-apples comparison, but can see that our description was a bit confusing. We reworked it to clarify for the reader what was done. We hope that this answers the reviewers request.
3. It makes sense to limit the implementation to leading-order for now, but given that almost all analyses at the LHC use next-to-leading order (or beyond) the authors must at least discuss the possibilities of extending their framework in that direction, and outline what the difficulties might be. It would also be interesting to understand if merged samples could be generated with Pepper. Our reply: Both the merged samples question and the possibilities of using GPUs in the calculation of one-loop matrix elements has now been added in the ‘Summary and Outlook’ section, in the revised first and newly added second paragraph.
4. This one doesn’t have to be addressed, but I just wanted to note that it isn’t clear that one needs to be on the ”native” branch (or download the native release) in order to compile without Kokkos. I managed in the end, although the code did not compile with my Cuda installation (Ubuntu + nvcc 12.3). After compiling I also could not find any documentation on how to run the code. I guess this will be added at a later stage? Our reply: • The introduction of the manual now has a prominent note explaining the separation into a native and a main/Kokkos version, cf. https://spice-mc.gitlab.io/pepper/intro.html • Our CI tests successfully tests compilation of the native branch with CUDA enabled using the “nvcr.io/nvidia/cuda:12.3.1-devel-ubuntu22.04” docker image, see https://gitlab.com/spice-mc/pepper/-/jobs/6180960798 for the latest compilation log. Hence we can not currently reproduce the reported compilation error. However, the CI builds have been added only recently, so perhaps the version that the reviewer tested has an issue which has been fixed in the meantime. We have improved and streamlined the getting-started tutorials for both the native and the main branch, see https://spice-mc.gitlab.io/pepper/tutorials/getting_started-native.html and https://spice-mc.gitlab.io/pepper/tutorials/getting_started-kokkos.html, respectively. • A guide for a first Pepper run is now added to the manual: https://spice-mc.gitlab.io/pepper/tutorials/running_pepper_for_the_first_time.html
Again, we would like to thank both referees for their insightful comments and thorough review of our work. We believe that this input has helped us to improve the draft significantly.
Best regards, Enrico Bothmann (on behalf of the authors)

---

## Round 3 · List of Changes

- Remove “first” in the abstract and the conclusions, and MadFlow references are now added in the introduction.
- Add ref. [3] in the introduction.
- In the introduction, mention now the developments of modern approaches to detector simulation and reconstruction, which is expected to reduce the relative cost of these steps in the simulation pipeline (and thus increase the
relative cost of event generation), and cite these approaches.
- Weaken the statement, in the conclusion, that parton-level generation will no longer contribute.
- Add a sentence in the introduction to stress that delivering a portable parton-level generator at LO is not only a natural starting point, but of immediate utility because tree-level matrix elements and phase-space sampling have the largest computational footprint even in state-of-the-art multi-jet merged calculations where the lower multiplicities are evaluated at NLO, as has been reported in [13, 39]. Further, we have acknowledged the importance of NLO calculations at the LHC in the Summary and Outlook, and discuss possibilities and challenges when extending Pepper towards NLO.
- Present validation at parton level in the new App. C. The appendix is referred to from the validation section. The new results show that the parton-level only data is also statistically compatible, directly comparing Pepper vs. Amegic without any further down-stream simulation steps.
- Add a second paragraph in the introduction of Sec. 5, to better define the scope of this work's performance studies, and refer to [32] for more technical studies.
- We have modified Sec. 3.1 to mention the possibility of dynamical scale choices. We have also noted that the strong coupling is also evaluated in parallel using the modified LHAPDF version.
- The fact that the scales can by evaluated dynamically is now also listed as step 5 of the new enumeration of the parallelized event generation steps in Sec. 3.3.
- We now refer to [39] in the Conclusions and Outlook, for a discussion and application for a multi-jet merged calculation. Therein, parton-level events by Pepper are read in by Sherpa, which perform the CKKW-type merging.
- Amend statement about a speed-up between our struct-of-arrays and array-of-structs layout when running on a CPU
- Add a dedicated study of struct-of-arrays and array-of-structs layout speeds on the CPU as App. D. Mention the main takeaways in the main text of Sec. 3.3, in order to keep the flow of the text intact.
- The event generation steps are now listed in Sec. 3.3. The question of what happens to events that do not pass phase-space cuts when evaluating many events in parallel is addressed below the list of steps.
- Clarify discussion of possible correlations between events.
- Add a short paragraph in the outlook discussing the precision requirements of our calculation, the issues that may arise from AI influences on hardware developments, and some approaches we are considering in addressing these concerns.
- Add a short discussion in the introduction about fully using HPC systems, which have a variety of architectures. Also, we highlight that providing a portable framework would enable the adoption of new more efficient hardware with minimal testing requirements.
- Clarify performance comparison discussion in second paragraph of Sec. 5.2.
- Add discussion points about merged samples and using GPU for one-loop matrix elements in the Summary and Outlook section, in the revised first and newly added second paragraph.

---

## Round 4 · Author Response

Dear referees,

Thank you again for your response and constructive comments.

We believe that we have addressed all remaining issues in our version four of the draft, which is available on arXiv as of today and which we have just now uploaded to SciPost as a resubmission. Furthermore, we have uploaded hardware reports created using Nvidia ncu and nvprof to Zenodo, and refer to them in the new draft.

In the following we will give our answers to each of the requested changes, which we repeat for easier reference.

A pdfdiff of the previous version with the new updated version can be downloaded here: https://www.theorie.physik.uni-goettingen.de/~bothmann/main-difffd5d8c855643a875308c1dcc6d630c7c18d80d4c.pdf

Requested change no. 1

Concerning the new Appendix D and Figure 10: As the author knows pretty well, a good usage of the hardware (M2 chip here) and a good memory layout, should lead to a factor of two speed up (speed-up of eight for the most modern HPC hardware). The authors do not observe such factor (either due to the memory layout, due to the code algorithm or to Kokkos itself or ...).

This weakens the "ecology" point of the paper, since it shows not fully efficient use of the hardware (but still better or on part to current used code). Additionally, if the issue is related to non-optimal memory layout, this should also impact GPU performance.

While fixing the issue might be too complicated for this paper, I think that that author should comment on this, on the probable cause and if they plan to fix it in the future.

Requested change: Comment on the fact that the code use poorly CPU+RAM paradigm (very minor)

Authors' reply

This is a misunderstanding caused by us due to not giving enough context in Appendix D/in the discussion of Figure 10, and perhaps by misunderstanding the point of the corresponding original question in the previous report of the referee. There is in fact very little vectorization at play here at all, as is now explained in the added footnote towards the beginning of Appendix D.

To remedy the fact that vectorization and CPU+RAM performance has been barely discussed so far, and to study the potential for further explicit vectorization across events, we have now added Appendix E. Here, we discuss the explicit vectorization implemented in the release version of Pepper (using the Vector Class Library of Agner Fog), which is restricted to vectorizing kinematic calculations of individual events, and we study the potential for implementing explicit vectorization across several events for the case study of the three-gluon vertex, using the Highway library, where we achieve very good speed-ups for various CPU vector sizes (of 1, 2, 4 or 8 doubles) with little modification of the released code (and no modification of the SoA layout of the data at all).

Requested change no. 2

Given the previous point, I want to reiterate, my previous request to see hardware reports showing how efficiently the hardware is used. While I understand that the authors do not want to do a technical paper, such information is crucial for the claim that portability is helping the environment. Those numbers can be provided as supplementary material online and might not even need to be formatted with text/... However, I will not block the publication if this is not provided.

Authors' reply

We published hardware reports created using Nvidia tooling (with ncu and nvprof), including Pepper-internal timing data, for the Z+jets and ttbar+jets processes studied in the paper on Zenodo, and refer to these datasets in the draft now (in the second paragraph of Sec. 5).

Requested change no. 3

The authors gave convincing argument about their strategy for their handling of the cuts but only as a reply to the referee. I think that the paper should include such arguments.

Authors' reply

This was not clearly communicated in our previous response. The discussion of the strategy towards cuts given in the reply is given in almost identical form after the enumeration of event generation steps in Sec. 3.3. It is the last paragraph of the section.

Again, we would like to thank you for your highly useful refereeing of our work. We believe that this input has further helped us to improve the draft significantly, which now contains extensive additional material and a lot of clarifications that the original version lacked.

Best regards, Enrico Bothmann (on behalf of the authors)

---

## Round 4 · List of Changes

- Add a footnote at the beginning of App. D to clarify that this appendix is not about CPU vectorization.
- Add App. E to discuss existing CPU vectorization in Pepper (to parallelize calculations of kinematic terms within single events) and to discuss a preliminary study on a more complete usage of CPU vectorization (to parallelize calculations across several events).
- Upload hardware reports to Zenodo as supplemental material. The reports are referred to in the second paragraph of Sec. 5.

---

## Editorial Decision

accepted_in_target_journal